# Bilayer Coating Systems: Functional Interlayers and Top Layers for Enhanced Performance

**DOI:** 10.3390/ma18225217

**Published:** 2025-11-18

**Authors:** Natalia A. Shapagina, Vladimir V. Dushik

**Affiliations:** Frumkin Institute of Physical Chemistry and Electrochemistry, Russian Academy of Sciences, Leninsky Prospect 31-4, 119071 Moscow, Russia; v.dushik@gmail.com

**Keywords:** functional sublayers, mechanical properties, corrosion resistance, duplex coatings, bilayer coatings

## Abstract

This review article summarizes the most widely used and effective technologies for producing protective and functional bilayer coatings. Particular attention is given to methods such as electroplating and electroless metallization, chemical vapor deposition, thermal spray and vacuum arc deposition, conversion treatments, laser modification, and organic layer deposition. Bilayer architectures are highlighted for their ability to overcome the limitations of single-layer coatings by combining complementary functionalities, resulting in enhanced adhesion, improved corrosion resistance through pore sealing or superhydrophobic surface states, and increased wear and crack resistance. This article is intended for researchers, materials scientists, and engineers engaged in surface engineering, corrosion protection, and advanced manufacturing, providing them with a clear understanding of the mechanisms, advantages, and practical applications of bilayer coatings. By synthesizing recent developments, comparative analyses, and performance data, the review enables readers to make informed decisions about the selection, design, and implementation of bilayer coatings for diverse industrial applications, ranging from aerospace and automotive components to medical devices and energy systems.

## 1. Introduction

Coatings have secured a prominent role among technologies that allow the targeted modification of material properties without the need to alter their bulk composition. In modern manufacturing, where high demands are placed on durability, wear resistance, corrosion resistance, and esthetic qualities, coatings serve as an effective tool for achieving the desired performance characteristics. Essentially, they enable the creation of material property combinations that are difficult or even impossible to attain through conventional fabrication methods. For this reason, the research and development of new coatings remain highly relevant and in demand across numerous industrial sectors, ranging from mechanical engineering and energy production to medicine and electronics.

A wide range of coating types exist, differing both in the nature of their base materials and in the properties they impart [1,2,3,4]. The following specification of coatings can be distinguished (Figure 1).

This classification reflects the diversity of objectives pursued in the application of coatings—from simple surface protection to the creation of functional layers with unique characteristics.

The methods for producing coatings are equally diverse (Figure 2).

They may be based on solid materials [5,6], as well as liquid [2,7,8] or gaseous media [9,10], which are transformed into a thin surface layer during the technological process. Various principles can be employed, including physical ones—such as spraying or condensation from vapor phase [10]—and chemical ones, such as electrochemical deposition [2] or chemical vapor deposition [9]. This variety of technological approaches makes it possible to flexibly select coating methods tailored to specific operating conditions and the desired coating properties.

However, the implementation of coatings also involves several limitations that cannot be overlooked. First and foremost, the mechanical compatibility between the coating and the substrate must be ensured. One of the most critical challenges is achieving reliable adhesion, without which a coating quickly loses its functionality. In addition, technological and economic constraints must be considered, including the complexity of the process, the cost of equipment, and consumable materials. Despite the wide range of existing coating techniques, few are sufficiently versatile to provide the full set of desired performance characteristics. The weak points of the “metal–coating” system often include adhesion strength, corrosion resistance, and crack resistance.

On the other hand, coating properties can be enhanced at relatively low cost and without significant complication of the process. This can be achieved by introducing an additional coating stage aimed at forming an intermediate sublayer beneath the main coating, or by improving the properties of the primary layer through post-treatment. The present paper focuses on a review of such two-stage coating technologies.

This study examines the following approaches to the formation of duplex coatings:Deposition of sublayers to improve adhesion, strength, and corrosion resistance using electroless and electroplating, application of oxide, phosphate or chromate conversion coatings;Surface modification of coatings through impregnation, hydrophobization, and vacuum-arc deposition to enhance corrosion resistance and tribological performance.

## 2. Functional Sublayers Deposited on Various Metallic Substrates to Improve Adhesion, Mechanical Strength, and Corrosion Resistance of Coatings

A functional sublayer is an intermediate layer of material applied to a substrate to improve the adhesion of the main coating. Such sublayers should exhibit strong adhesion to the substrate, thermal stability, compatibility with the top coating, and resistance to aggressive environments. When these requirements are met, the coating deposited on the functional sublayer gains several advantages: enhanced adhesion strength to the metal, extended service life, reduced risk of delamination, and improved performance in corrosive conditions. Typically, epoxy compositions, polyolefin systems, and metal-containing layers are used as sublayers. This section provides a detailed discussion of alternative types of functional interlayers and the methods used to form them on various metallic substrates (Figure 3).

When choosing the method for applying the upper sealed and functional layers, it is important to consider the characteristics of the metal base—the possibility of creating a two-layer coating depends on these.

### 2.1. Ni–P Support Layers for Chemical Vapor Deposition (CVD) W–C Coatings

The most common industrial methods for applying metallic and ceramic coatings include cladding, electroplating, anodizing, as well as thermal spray and vacuum deposition techniques. These methods are characterized by high productivity and technological efficiency and can be used for a wide range of materials [9,10,11,12,13]. However, their effectiveness decreases when processing components with complex surface geometries. This limitation arises from geometric factors affecting the distribution of electric field forces or from the size and limited mobility of spraying equipment.

An alternative in such cases is the CVD method. Its principle is based on the chemical transformation of gaseous reactants on the surface of a substrate, resulting in the formation of metallic or metal–ceramic coatings [9,12,14]. The advantages of the fluoride-based CVD technology include the relatively low process temperature, high penetration capability, and strong chemical activity of the gaseous medium, which make it possible to produce dense and uniform layers even on components with complex geometries, without the need for plasma or external physical fields [15,16,17,18]. Nevertheless, fluoride-based CVD is not without its drawbacks. The main challenges include the high cost of the process and poor adhesion of coatings to iron-containing alloys, particularly steels of all industrial grades. To overcome this issue, intermediate nickel adhesion layers are commonly used, providing a strong bond for coatings deposited from fluoride-containing media [19,20].

The method for depositing a nickel sublayer must also take advantage of CVD while remaining insensitive to the shape of the component. Using CVD for this purpose involves highly toxic compounds, such as nickel tetracarbonyl Ni(CO)_4_ [14]. This increases the safety requirements for production facilities and simultaneously leads to contamination of the reaction zone with oxygen-containing decomposition products from the reaction Ni(CO)_4_ → Ni + 4CO, which hinder the growth of coatings from tungsten hexafluoride.

A safer and more technologically flexible alternative is electroless plating [21,22]. This method allows for the formation of coatings from nickel alloys of various compositions, does not require external current sources, and like CVD, is largely insensitive to the complexity of the component geometry. The fluoride-based process for depositing tungsten and its carbides offers an advantage over other CVD technologies, as it enables the formation of hard coatings on structural materials at relatively low temperatures (450 ÷ 600 °C) [10,15]. However, in addition to the aforementioned adhesion issues of such coatings to steels, another problem arises due to the mechanical mismatch between these materials. Studies [23,24,25] suggest that surface layers with properties that gradually change through their thickness exhibit the best mechanical performance. Not all methods, however, allow for the creation of gradient coatings, making it practical to aim for a stepwise variation of properties. Thus, when depositing hard coatings with hardness up to 4000 HV on relatively soft materials with hardnesses up to 300 HV, it is advisable to form an intermediate sublayer. This sublayer should possess intermediate hardness and provide strong adhesion of the top coating to the substrate, effectively serving as a support layer.

The electroless plating method is known for its ability to produce nickel alloys with phosphorus, boron, and certain metals [7,21,22], which may not initially exhibit high hardness. However, it is well-established that vacuum annealing can increase the hardness of such coatings. Figure 4 shows the effect of 1 h vacuum heat treatment on the hardness of Ni–(W,Mo)–P(B) chemical coatings. It is evident that annealing significantly enhances their hardness, and the optimal temperature range of 300 ÷ 600 °C coincides with the ideal deposition temperatures for CVD WC coatings.

The authors of [26] investigated the potential of using electroless plating to create a support sublayer for hard CVD WC coatings. It was shown that Ni–(W)–P layers provide good support properties, resulting in increased adhesion and cohesion strength of the coatings, as well as an increase in the microhardness of the intermediate layer from 100 ÷ 200 HV to 500 ÷ 700 HV, particularly compared to metallurgical nickel as a substrate or an electroplated intermediate nickel layer [27].

This approach is based on the idea that the properties of the intermediate layer can change during the deposition of the top coating if the process involves elevated temperatures. Therefore, the intermediate layer and its deposition method should be selected so that these changes contribute to the improvement of the mechanical properties of the substrate–intermediate layer–coating system.

### 2.2. Conversion Coatings

Conversion coatings are protective layers formed as a result of a chemical reaction occurring on the surface of a metal. The most commonly used types of conversion coatings in industry include phosphate, chromate, and oxide coatings [28].

#### 2.2.1. Phosphate Conversion Coatings (PCCs)

Phosphating is a method used to produce PCCs on components made of various structural metals, including ferrous materials (carbon and low-alloy steels, cast iron) and non-ferrous metals (aluminum and its alloys, zinc, magnesium, cadmium, and copper-based alloys). The method is based on the ability of metals to form insoluble phosphate compounds on their surface through interaction with phosphoric acid solutions and their salts, resulting in a phosphate film [29].

Phosphate conversion coatings exhibit strong adhesion to the metal surface. For example, when a phosphated steel sheet is bent, the coating may crack or fracture, but it does not delaminate. The plate-like phosphate crystals form a well-developed microporous structure, allowing the film to effectively absorb and retain lacquers, paints, and lubricants. The phosphate film also possesses high electrical insulation properties. The dielectric strength can reach up to 1000 V and may be further increased by impregnation with special insulating varnishes. Thermal stability and insulating properties are maintained up to 825 ÷ 875 K.

The thickness and structure of PCCs depend on the composition of the metal, the phosphating conditions, and the surface preparation method. As a result, the film thickness can vary widely—from fractions of a micrometer to more than 100 µm. The color of PCCs ranges from light gray to dark gray (almost black). Light-gray coatings are typically observed on non-ferrous metals and soft steels that have undergone sandblasting or treatment in highly acidic solutions [29,30,31,32,33,34].

PCCs can exhibit either crystalline or amorphous structures. For instance, in [32], a method for phosphating aluminum surfaces was proposed that involved adding fluorine compounds together with primary phosphates of manganese, zinc, and cadmium, or incorporating chromic acid into the solution. As a result, a two-layer PCC was formed on the aluminum surface: the first, near-surface layer, is thin (up to 1 µm), smooth, elastic, and highly porous; the second, outer layer, determines the overall properties of the coating, has a crystalline structure, and consists of secondary and tertiary phosphates. As the outer layer grows, the metal surface becomes isolated from the solution, gradually reducing the reaction rate. The resulting phosphate coatings are crystalline films composed of water-insoluble phosphate salts.

The formation of amorphous PCCs on copper-containing aluminum alloys D1 and D16 was investigated in [33]. To accelerate phosphating, fluorine and molybdenum (IV) compounds were used. X-ray photoelectron spectroscopy revealed that the formed PCCs consisted of aluminum oxides, phosphates of the metallic alloy components, and molybdenum (IV, V) phosphates.

Various types of phosphating can be distinguished for forming phosphate conversion coatings (PCCs) on different structural metals:

Chemical phosphating—used for metals with relatively weak structures such as aluminum, low-alloy steel, magnesium, and zinc. A subtype of this process is amorphous phosphating, which employs iron phosphates [30,31].

Accelerated phosphating—performed in solutions containing nitrates. For instance, the introduction of 50 ÷ 70 g/L of zinc nitrate into a solution with increased acidity and the presence of a manganese–iron phosphate formulation allows for a reduction in the treatment time to 10 ÷ 20 min. This method is suitable for critical components such as springs made from 0.5 mm wire [32,34,35,36].

Black phosphating—used mainly for the decorative treatment of metallic parts, resulting in a black surface film with sufficient density to increase mechanical strength [32,34].

Zinc phosphating—applied to zinc and its alloys. Two main types of solutions are used: zinc-containing and zinc-free. Zinc phosphating provides the best protective effect, although it is more expensive than other techniques [32,36].

Cold phosphating—the most economical method since the solution remains stable for a long time and does not require additional safety measures (e.g., local ventilation or tank covers). This technique is often used for spray coating and for repairing or restoring coatings in service conditions [32,35].

Hot phosphating—commonly used in industrial practice. For example, coatings containing iron and manganese salts can be obtained from a solution containing 20 ÷ 40 g/L of a manganese–iron phosphate formulation, without stirring, at 92 ÷ 98 °C for 1 h [36,37].

Normal phosphating—the best results are achieved using a solution containing 30 ÷ 33 g/L of a manganese–iron phosphate formulation at 98 °C. At higher temperatures, excessive sludge forms, whereas at lower temperatures, the coating becomes overly crystalline. The total process time (hydrogen evolution and holding) is about 5 ÷ 10 min [36].

Electrochemical phosphating—performed using either alternating or direct current. The resulting film is used as a primer for subsequent painting. In electrochemical phosphating with alternating current, manganese–iron phosphate or accelerated phosphating solutions are used as electrolytes. The applied voltage is typically 15 ÷ 20 V, and the workpieces themselves serve as electrodes [37].

Due to their microporous structure, PCCs alone do not provide reliable corrosion protection. They are stable under atmospheric conditions, in lubricating oils and organic solvents, but degrade in acids and alkalis. Therefore, PCCs are usually used as a base for paints or polymer coatings, which ensure good adhesion to steel [34,38]. In some cases, PCCs are impregnated with oils, paraffin, or treated in diluted solutions of chromic, phosphoric, or oxalic acids, iron and chromium nitrates, liquid glass, or triethanolamine, which significantly increases their corrosion resistance [29,39,40].

Currently, another well-studied method for forming conversion coatings is chromating, which enables the formation of films with not only good decorative appearance but also enhanced corrosion resistance.

#### 2.2.2. Chromate Conversion Coatings (CCCs)

Chromate conversion (passivation) is a process in which a metallic surface is treated by immersion in a solution of chromic acid and its salts to form a thin oxide film with corrosion-resistant properties. CCCs are used for the passivation of aluminum, zinc, steel, cadmium, copper, silver, titanium, magnesium, and tin alloys. The process of applying CCCs to a metal surface can be divided into several stages: cleaning the substrate from dirt and oils; immersing the metal in a chromium-based solution to form a protective film; rinsing off excess solution to ensure uniform coverage; and drying the metal to complete the process [41,42,43,44,45].

Freshly deposited CCCs are generally amorphous, gel-like, and soft, exhibiting low mechanical strength and requiring careful handling. Upon drying, they shrink and harden, become difficult to wet, and resistant to aqueous solutions. The curing process continues for up to 24 h after drying [46]. The quality of CCCs is influenced by many factors, including the composition, concentration, and acidity of the chromating solution; the quality of the metallic substrate (presence of impurities, structure, gloss, roughness); treatment and rinsing duration; chromating bath temperature and post-rinse temperature; and drying temperature. These factors can significantly affect the physicomechanical properties of CCCs, such as:Porosity—CCCs are porous before drying. Thicker coatings generally contain fewer pores.Hardness—higher chromating solution temperatures produce harder coatings;Thickness—determined by the solution composition, particularly pH. Lower pH values result in thicker coatings. Thick CCCs are less wear-resistant initially, but their wear resistance increases upon drying [44].Adhesion and plasticity—CCCs are generally sufficiently plastic, and cracks formed during deformation can self-heal to some extent [46].

CCCs are used in industries where metal components are exposed to harsh conditions. In the aerospace industry, these coatings protect aluminum and metal alloys from corrosion, being applied to components such as the chassis and engine mounts to ensure safety during flight. In the automotive industry, CCCs protect steel and galvanized parts from weathering, extending the service life of vehicles [47].

Two main types of chromating can be distinguished for forming CCCs on various structural metals:Transparent CCCs—formed using trivalent chromium compounds. These coatings allow the natural color of the metal to show through, making them suitable for industries focused on appearance, such as automotive and electronics. They provide corrosion resistance and are suitable for outdoor and marine applications [48,49,50,51].Yellow CCCs—formed using hexavalent chromium compounds, which impart a bright yellow color to the metal. These coatings are used for decorative purposes, such as in plumbing and equipment, and offer superior corrosion resistance, making them versatile for various applications. The formation of transparent CCCs is considered more environmentally friendly compared to yellow CCCs.

Despite advances in conversion coatings, CCCs formed with trivalent chromium compounds have not fully replaced hexavalent chromium CCCs. Due to the absence of a self-healing effect, the most vulnerable areas of trivalent chromium CCCs are sharp edges, threads, and crevices [52,53,54].

Though CCCs remain highly effective in terms of protection but are highly toxic, finding other ways of applying anticorrosive coatings is a relevant purpose. Taking into considerations the environmental agenda, the following options for applying bilayer protective coatings can be used:Change to phosphate and oxide (chemical) conversion coatings in comparison with CCCs and less toxic substances are used to apply phosphate and oxide-conversion coatings. A moderate level of energy consumption characterizes these coating methods;Use other anti-corrosion coating technologies: electroless and electroplating, chemical heat treatment, hydrophobization, and vacuum arc spraying. However, in most cases, these coating methods consume a lot of energy;Create recyclable coatings. Here, it is possible to use polymer materials suitable for recycling; the development of formulations with the possibility of recovery; the use of biodegradable components in some types of coatings.

The selection of a specific coating type should be based on a comprehensive analysis of protection requirements, operating conditions, and environmental restrictions.

#### 2.2.3. Oxide-Conversion Coatings (OCCs)

Oxidation is the process of forming a thin oxide film on the surface of a metal product to improve its protective and decorative properties. Oxide coatings (OCs) can be obtained by various methods:Thermal—oxidation of metal by oxygen in the air at moderately high temperatures. The process can be carried out in air, in a water vapor environment, in mineral or vegetable oils, or in molten salts [36,55].Electrochemical—oxidation by oxygen generated on the metal surface as a result of an electrochemical process. Usually, products are treated at the anode in an alkaline solution with added oxidizers. Due to the complexity of the process, this method has found limited application [55].Chemical—oxidation of metal in liquid media. This type of oxidation is widely used to obtain protective and decorative coatings of black or dark blue color. The process is carried out in concentrated caustic solutions with added oxidizers such as sodium or potassium nitrates. The protective ability of such OCs is significantly increased if phosphoric acid and nitrates of certain metals are added to the solution [55].

Both ferrous (steel) and non-ferrous metals (copper, zinc, aluminum, titanium, magnesium, and their alloys) can be oxidized. The process of forming OC and the properties of the resulting coatings largely depend on the substrate material:Steel (steel blackening)—a process of forming an iron oxide film 1 ÷ 10 µm thick on the surface of steel and cast iron. Stainless steels are not oxidized, while alloyed and high-alloy steels are more difficult to oxidize. Types of blackening include: alkaline—conducted in alkaline solutions with oxidizers at 135 ÷ 150 °C; acidic—conducted in acidic solutions chemically or electrochemically [55]. Since alkaline bluing uses caustic solutions at temperatures significantly above the boiling point of water, and the process is long, it is difficult to call it convenient or environmentally friendly. Therefore, research continues to improve steel blackening methods. Iron dissolution rate depends on chemical composition and microstructure: high-carbon steels oxidize faster than low-carbon steels. Steel composition affects the OC color: low-carbon steels produce deep black coatings, while high-carbon steels yield black with a grayish tint. OC on steel has a fine-crystalline, microporous structure. To enhance shine and protective properties, the coating is impregnated with oil (mineral or vegetable), which fills the pores, improving corrosion resistance, wear resistance, and deepening the black color. Oiled OCs on steel are used for corrosion protection, decorative finishing, and as an anti-glare coating on tools [56,57,58].Copper—highly resistant to corrosion due to its position in the electrochemical series. Oxide-conversion coatings on copper and its alloys are used for blackening, increasing the light absorption of optical components, decorative finishing, and improving adhesion for bonding. Copper OCs can be obtained by:Thermal (hot oxidation)—copper and its alloys are heated in the presence of an oxidizer (e.g., molten nitrate or oxygen-rich atmosphere). Temperature and time determine the thickness and color of the thermal OC: higher temperatures produce thicker, darker coatings. This method forms thick, durable OCs but requires high temperatures and precise control [34,36,59,60].Electrochemical (anodizing)—copper or copper alloy acts as the anode. Electrolytes are usually alkaline solutions, most often sodium hydroxide. Current density, voltage, temperature, and duration determine color and thickness. For example, in 0.25 N NaOH at 40 °C, current density 1 ÷ 2 A/dm^2^, and 10 min, light-colored anodized OC forms; 2 ÷ 4 A/dm^2^ produces darker, nearly blue coatings. Increasing temperature to 60 °C expands the current density range for dark shades to 2 ÷ 6 A/dm^2^. At 2 ÷ 3 A/dm^2^, brown films with bluish-green tints can be obtained in NaOH solutions of 0.25 ÷ 1.0 N. Anodized OCs have good adhesion and corrosion resistance [37,61,62,63].Chemical oxidation—copper is treated with solutions containing oxidizers, commonly persulfate or copper-ammonia solutions. Persulfate gives black coatings superior mechanical and anti-corrosion properties compared to copper-ammonia solutions. Alloys with less than 90% copper may require lower persulfate concentration or pre-coppering. Copper alloys with 50 ÷ 65% copper produce black films with a bluish tint, 2 ÷ 3 µm thick [60,64,65,66,67]. Chemical oxidation is simpler than electrochemical, but coating quality depends on solution composition and treatment conditions. Post-treatment with oil, wax, or lacquer improves corrosion resistance and appearance. The choice of method depends on coating requirements and production conditions.Zinc is a chemically reactive metal. Under conditions of high humidity and in chemically aggressive environments, zinc coatings corrode relatively rapidly, which inevitably deteriorates the appearance of the product. In this regard, the formation of an oxide-conversion coating (OCC) on the zinc surface significantly slows down the corrosion process, improves the adhesion of paint and varnish materials, and provides a decorative appearance to the zinc product (serving as one of the methods for blackening zinc). There are two main approaches to zinc oxidation. The thermal method involves cleaning and degreasing the surface, followed by treatment with a solution composed of equal parts of 25% copper acetate and 30% acetic acid. The sample is then heated to 300 °C for 2 min, and the procedure is repeated twice. The chemical method, in turn, employs a solution with the following composition (g/L): phosphoric acid, 2 ÷ 10; sodium nitrate, 70 ÷ 100. The treatment duration for zinc products is 30 ÷ 40 min at a temperature of 80 ÷ 100 °C. As a result of these processes, a smooth and matte OCC with good electrical insulating properties and a thickness of up to 40 µm is formed on the zinc surface. To further enhance corrosion resistance, it is recommended to carry out an additional treatment with oil [37,68,69,70].Aluminum is also known as a highly reactive metal; however, its surface is naturally covered by a protective passive film of Al_2_O_3_ with a thickness of 2 ÷ 5 nm, which significantly retards corrosion processes under atmospheric conditions. Nevertheless, due to its limited thickness, this oxide film does not provide sufficient corrosion resistance or adequate physical and mechanical properties in aggressive environments. To improve these characteristics, the thickness of the oxide film must be artificially increased, that is, through oxidation. In this context, three main oxidation methods are employed to form oxide-conversion coatings (OCCs) on aluminum surfaces: chemical oxidation, anodizing, and plasma electrolytic oxidation (PEO) [36,37,71,72,73,74,75,76,77,78,79,80,81,82,83,84,85,86].Titanium belongs to the class of transition metals and exhibits remarkable stability in many environments, maintaining its resistance at room temperature and in air up to 550 °C. This corrosion resistance is attributed to the presence of a thin but dense oxide film on its surface. The thickness of this film reaches 5 ÷ 20 nm, which is slightly greater than that of aluminum, but is significantly stronger on titanium. The natural oxide layer on titanium primarily consists of rutile (β–TiO_2_) and anatase (α–TiO_2_). At temperatures above 600 °C, titanium actively reacts with oxygen, forming pure rutile. To enhance the protective capability of the natural oxide film on titanium, as well as improve its antifriction and physicomechanical properties, anodizing or PEO processes are employed [87,88,89,90,91,92,93,94].Magnesium is the eighth most abundant element on Earth, providing ample resources for the use of Mg and its alloys across various engineering sectors. The advantages of magnesium alloys include high strength, low weight, and non-toxicity to both the environment and the human body [95,96,97,98,99,100]. However, magnesium possesses a highly electronegative electrode potential and exhibits a poor protective capability of its surface films due to their defective nature [99,100,101,102,103,104]. To ensure effective protection of magnesium and its alloys, chemical oxidation formulations have been developed, in which the main components are chromium salt compounds. As a result, oxychromate oxide-conversion coatings (OCCs) with a thickness of several micrometers are formed on the magnesium surface, with their color depending on the composition of the solution and the alloy used [105]. Due to the toxicity of such solutions, alternative methods have been developed to improve the corrosion resistance of oxide layers on magnesium parts and components. Currently, anodizing, the Dow-17 process, and PEO are the most widely employed techniques [98,100,106,107,108,109,110,111,112,113,114,115,116].

Chemical Oxidation of Aluminum

Chemically formed oxide-conversion coatings (OCCs) on aluminum typically have a thickness of 0.5 ÷ 4 µm and a microporous structure, which provides high adsorption capacity and consequently, improved adhesion of paint coatings to the surface of components. Although such OCCs are inferior to anodic and PEO coatings in terms of performance characteristics, they offer significant technological and economic advantages when coating complex-shaped or large-sized parts, internal surfaces of long and thin-walled tubes, large welded structures, as well as in applications requiring electrical conductivity on the oxidized aluminum surface. Chemical OCCs on aluminum and its alloys are generally produced from alkaline–chromate, phosphate–chromate, and chromate–fluoride solutions. Alkaline–chromate OCCs are less than 2 µm thick and possess limited mechanical properties; thus, they are mainly used as primers for subsequent painting. Phosphate–chromate OCCs have greater thicknesses (up to 4 µm) and exhibit improved protective and physico-mechanical properties, making them suitable for both corrosion protection and as primer layers. Chromate–fluoride OCCs are thin but dense and are characterized by low electrical resistance, which makes them suitable for forming electrically conductive oxide coatings [117,118,119]. As previously mentioned, the use of chromate-containing solutions no longer meets modern environmental and safety requirements; therefore, researchers continue to develop technologies for forming OCCs on aluminum and its alloys without the use of toxic chromates [120,121,122,123,124].

Anodizing Aluminum, Titanium, MagnesiumAluminum

The most widespread method of oxidizing aluminum surfaces is anodizing. Anodizing can be regarded as a specialized technological process within the broader phenomenon of electrolysis, aimed at achieving a specific result—the formation of a protective oxide film on a metallic surface [75,76,77,78,79].

Anodic OCCs are formed electrochemically on aluminum surfaces from aqueous acid or alkaline solutions. During electrochemical anodizing, both mildly aggressive electrolytes (phosphoric, citric, and boric acids) and highly aggressive ones (oxalic, sulfuric, sulfosalicylic acids, and chromic anhydride) are used. Anodizing is generally carried out at elevated voltages, ranging from 24 to 120 V, depending on the electrolyte. As current passes through the electrolyte, depending on its composition, the reaction products on the aluminum anode may: (1) dissolve completely without forming a coating; (2) produce a compact, adherent, and electrically insulating OCC with a thickness of tens of nanometers; or (3) partially dissolve to form a porous OCC with a thickness of tens to hundreds of micrometers. After formation, a porous coating may remain unchanged, be sealed in water, or be impregnated. In the first case, the coating serves as an excellent substrate for paint and adhesive applications; in the second, it retains a silvery appearance and gains higher corrosion resistance; in the third, it can be colored without the need for paint coatings [125,126,127].

Two primary theories describe the formation and growth of anodic coatings: the structural–geometric (Keller’s cell model) and the colloidal–electrochemical (Bogoyavlensky’s theory) [87,128,129,130,131,132,133]. According to the first theory, when anodic voltage is applied to an aluminum electrode (i.e., connected to the positive pole), a compact oxide film initially forms. In electrolytes that dissolve oxides, the outer part of this film begins to dissolve at defect sites, leading to the development of a porous structure. Further coating growth occurs at the base of these pores through the oxidation of deeper metal layers. The coating consists of hexagonal cells. The barrier layer adjacent to the metal, about 1.1 nm thick, is nonporous, while the porous layer consists of cells each containing a central pore. The pore size and density depend on the electrolyte composition and anodizing conditions. Under the influence of the electrolyte, the oxide forming the cell walls becomes hydrated, involving the adsorption of water, anions, and reaction products [87,128,133].

From the perspective of the second theory, the formation of anodic coatings begins with the appearance of minute oxide particles produced by the collision of ionic fluxes. Adsorption of anions and water gives these particles a negative charge. As their number increases, they combine into polyions—rod-shaped micelles—that form the framework of an oriented aluminum oxide gel. Anions from the electrolyte penetrate this gel. Driven by their negative charge, the micelles migrate toward the surface and fuse with the metal. Alongside micelle formation, dissolution processes of the nascent oxide occur concurrently [129,130,131,132].

Anodic OCCs can be either thin, nonporous coatings or thick, porous ones. Thin, nonporous anodic OCCs are obtained from mildly aggressive electrolytes and consist primarily of anhydrous aluminum oxide located near the metal interface. At the oxide/electrolyte interface, a small fraction of hydrated aluminum oxide (boehmite, Al_2_O_3_ × H_2_O) is found. Due to the absence or scarcity of pores, such coatings have limited dyeability [131,132,133]. Thick, porous anodic OCCs are formed in aggressive electrolytes and consist of amorphous aluminum oxide, partly containing γ–Al_2_O_3_. The water content can reach up to 15%. Depending on the formation conditions, water may exist as part of boehmite or bayerite (Al_2_O_3_ × 3H_2_O). These coatings comprise two layers: a barrier layer, adjacent to the metal surface, nonporous and 10 ÷ 30 µm thick; and a porous layer, a system of conical pores penetrating the oxide film, with thicknesses ranging from several micrometers to millimeters. The coatings may contain up to 20% electrolyte anions [87,131,132,133].

Anodic OCCs on aluminum can be produced by several methods, each with its own advantages and limitations. Coating properties depend on current density and electrolyte temperature: higher current densities and lower temperatures result in harder anodic layers.

Hard anodizing involves electrolytes composed of acid mixtures—sulfuric acid combined with oxalic, acetic, boric, or orthophosphoric acids, chromic trioxide, and various organic compounds. The electrolyte temperature is typically 20 ÷ 30 °C. This method is widely used in modern industry to form thin, durable anodic OCCs on aluminum and its alloys [134,135,136,137].Warm anodizing is performed at 15 ÷ 20 °C. Aluminum is treated until a light milky film forms, then rinsed with cold water and dyed with aniline-based solutions. This process produces esthetically pleasing surfaces but provides limited protection under harsh conditions, offering lower corrosion, chemical, and mechanical resistance. Such coatings, however, serve well as substrates for paint finishes [138,139,140,141].Cold anodizing is carried out at −10 to +10 °C. Coatings produced under prolonged forced cooling form dense films, which are then sealed with steam or hot distilled water. This method yields high-quality, thick, and durable coatings [87,142,143,144].

Regardless of the method, anodized OCCs significantly enhance the corrosion and physico-mechanical properties of aluminum and its alloys compared to bare metal. They exhibit excellent resistance in marine atmospheres and seawater, reduce aluminum corrosion in acetylene, sulfur dioxide, boric acid, benzene sulfonic acid, ethanol, and ethanol solutions. In humid environments, pore wall hydration leads to the formation of boehmite or hydrargillite, increasing the coating weight, compactness, and corrosion resistance over time. In chloride-containing media, corrosion proceeds locally along pores, producing variable-composition aluminum hydroxychlorides that gradually transform into hydroxides, sealing pores and slowing corrosion.

The microhardness of anodic OCCs depends on the substrate composition (in GPa): up to 5.1 for pure aluminum; 4.9 for alloy AB; 4.7 for alloys of the AL series; and 3.6 for alloy D16. To improve strength and electrical insulation, thick coatings (typically 40 ÷ 90 µm, occasionally several tenths of a millimeter) are produced. Such coatings exhibit high thermal resistance, withstanding temperatures up to 2000 °C, which makes anodized aluminum suitable for molds used in casting aluminum and magnesium alloys [75,76,77,78,79,87,128,133,134,135,136,137,138,139,140,141,142,143,144].

The ability of anodic OCCs to adsorb organic compounds underlies the process of dyeing. Coatings obtained in oxalate electrolytes usually have a yellowish tint. When aluminum and its alloys are anodized, first under alternating current and then direct current in such electrolytes, coatings appear in shades from light straw to golden and bronze. Transparent and translucent protective–decorative coatings can be dyed using aqueous solutions of acid organic dyes. Dye concentration ranges, g/L: 0.1 ÷ 0.5 for light tones; up to 5 for intense shades; and 10 ÷ 15 for black color, at 50 ÷ 70 °C and dyeing times of 5 ÷ 30 min. Coatings formed in different electrolytes vary in color due to differences in porosity and inherent film color. Desired hues are achieved using mixtures of aniline dyes. Poor-quality coloring can be removed in potassium permanganate and nitric acid solutions. In addition to organic dyes, inorganic ones are also used—these produce a more limited color range but superior lightfastness through ion-exchange reactions in inorganic salt solutions [36,37,87,133].

Aluminum and its alloys after anodizing have found broad applications across numerous industries due to their universal nature, durability under various environmental conditions, and appealing esthetic appearance.

Titanium

The process of forming anodic oxide-conversion coatings on titanium and its alloys involves immersing the metallic component in an electrolyte and connecting it to the positive pole of a direct current source, while the cathodes are connected to the negative pole. During the passage of electric current through the electrolyte, oxygen is released at the anode in its active form, which reacts with titanium to form an anodic oxide-conversion coating. The growth of this coating does not occur on the external surface of the part but beneath the previously formed oxide layer, i.e., at the interface between the titanium substrate and the anodic film. Typically, aqueous electrolytes of sulfuric, phosphoric, or oxalic acid, as well as combined compositions of sulfuric and phosphoric acids in pulse mode, are used for the electrochemical anodizing of titanium [36,37,88,91].

Anodizing in sulfuric acid is performed in two modes:Stationary mode—anodizing is carried out at a constant current density of 1 ÷ 1.5 A/dm^2^ with voltage increasing from 5 to 25 V. The resulting anodic coating has a thickness of 6 ÷ 12 µm.Pulse mode—the direct current source provides short pulses of 0.1 ÷ 0.3 s at a frequency of 120 pulses per minute, exceeding the working current by 5 ÷ 8 times. To obtain a coating thickness of 15 ÷ 20 µm, a current source capable of providing up to 50 A/dm^2^ is required. At the end of the process, the voltage rises to 250 V, and bath cooling is necessary. This procedure produces layer-by-layer compaction of the coating, resulting in low porosity. Coatings obtained from sulfuric acid electrolytes have a specific electrical resistance of σ = 3.7 × 10^−8^ Ω × cm [88,145,146,147,148,149].

Anodizing in phosphoric acid is carried out at low current densities of 0.2 ÷ 0.8 A/dm^2^ and voltages of 50 ÷ 150 V. This allows for the formation of colored decorative anodic coatings without using additional dyes, with thicknesses up to 10 µm. The coating color depends on the applied voltage and the substrate material (titanium alloy grade), with possible shades including brown-yellow, blue, light blue, yellow, pink, crimson, green, and various tones. Color variations arise from differences in the crystalline structure of the anodic layer. The thickness of the oxide layer affects the refraction of incident light at different angles, subtly changing the perceived color. This mechanism differs from aluminum anodizing, where dyes determine the color independently of the viewing angle. Decorative anodizing of titanium alloys is used for esthetic purposes and marking, increasing corrosion resistance and lightfastness while maintaining the metallic surface’s natural gloss. The widest color range and highest saturation have been achieved on VT-20 (pseudo–α alloy) and VT-6 (α+β alloy). In α+β alloys containing molybdenum and chromium, the quality of anodic films is lower, and commercially pure titanium exhibits a limited color range. Titanium alloys containing manganese (OT4, OT4-1) are not recommended for decorative anodizing [87,88,150,151,152,153].

Anodizing in oxalic acid is performed at a constant voltage of 110 V and current densities of 1 ÷ 3 A/dm^2^. The resulting oxide layer, 2 ÷ 4 µm thick, improves the tribological properties of titanium and its alloys [87,88].

Pulse anodizing in combined electrolytes involves current pulses of 0.1 ÷ 0.3 s with a frequency of 60 ÷ 120 pulses per minute and pulse current densities of 5 ÷ 10 A/dm^2^. The coating thickness and quality depend on the ratio of pulse duration to pause duration. At a current density of 1 ÷ 2 A/dm^2^, coatings reach up to 3 µm in thickness, while powerful power supplies allow for coatings up to 20 µm thick (current densities up to 50 A/dm^2^) [88,154,155].

The mechanism of oxide layer formation and growth is determined by the chemical composition and crystalline structure of the substrate, the type of electrolyte, and the technological parameters of the process. Surface roughness is also a critical factor: rough surfaces allow for the formation of thicker, more adherent coatings with higher open porosity, while on smooth surfaces, thick coatings are prone to delamination and failure.

Magnesium

During the passage of electric current through an electrolyte, the surface of magnesium undergoes oxidation, resulting in the formation of an anodic oxide-conversion coating. The magnesium component acts as the anode, while cathodes can be made of steel, graphite, platinum, or other materials. The process is carried out at a constant current density, the maximum value of which is limited by the onset of pitting on the anode. Under the influence of the current, magnesium ions migrate into the solution, while an oxide layer forms on the surface.

For anodic oxidation of magnesium alloys, alkaline solutions with the addition of substances that oxidize at the anode, such as silicates, borates, and phosphates, have been proposed. These substances react with magnesium to form insoluble compounds. During the anodizing process, the solution temperature is maintained within 20 ÷ 80 °C, the duration ranges from 20 to 45 min, and the resulting anodic coatings have a thickness of 5 ÷ 25 µm. The color of the anodized coatings varies depending on the alloy composition, ranging from dull green to glossy black. Higher aluminum content in the alloy results in a darker coating. The thickness and structure of the oxide layer depend on the anodizing conditions, including the electrolyte composition, temperature, current density, and processing time.

Anodized coatings on magnesium provide high corrosion resistance against moisture and oxygen, increased surface hardness, and improved resistance to mechanical damage. They also enhance adhesion with paints and other coatings while maintaining the lightweight nature of the material. To preserve the protective properties, anodized magnesium requires regular inspection for damage, cleaning with soft non-abrasive agents, and avoidance of aggressive cleaning chemicals. Anodizing magnesium is therefore an effective method to enhance the operational performance of this lightweight and strong metal, making it highly valuable in aerospace and automotive industries, electronics, sports equipment, and medical applications [156,157,158,159,160,161,162].

A specialized variant of magnesium anodizing is the Dow-17 process, which produces a hard, corrosion-resistant coating applicable to all magnesium components regardless of the alloy. The Dow Chemical Company developed the Dow-17 technology, the first commercial anodic coating for magnesium, in the mid-1940s, when the company was the largest magnesium producer in the world. Dow-17 can be applied using either alternating or direct current. The electrolyte consists of sodium dichromate, ammonium fluoride, and phosphoric acid, with a pH of approximately 5, and the operating temperature must be at least 70 °C. As a result of anodizing in this electrolyte, a green or dark green coating forms on the magnesium surface, with the exact shade depending on the alloy. After the Dow-17 process, the corrosion and wear resistance of components are significantly improved, and the surface is ready for subsequent processing, including painting. Therefore, Dow-17 represents a critical process for enhancing the durability and reliability of magnesium components [163].

Plasma Electrolytic Oxidation (PEO) of Aluminum, Titanium, Magnesium

PEO is a relatively recent method for the surface modification of aluminum, developed at the Institute of Inorganic Chemistry of the Siberian Branch of the Russian Academy of Sciences in 1969 under the supervision of G.A. Markov. This technique enables the deposition of ultra-durable OCCs with unique protective, electrical insulating, and decorative properties. Visually, PEO coatings resemble ceramics. The process is applicable not only to aluminum but also to other valve metals, such as titanium, magnesium, zirconium, tantalum, and beryllium [164,165,166,167,168,169,170,171,172,173,174].

PEO is conducted in an electrolyte under the passage of electric current, similar to anodizing, but differs in the use of significantly higher voltages and high-current densities. When such a current passes through the metal/electrolyte interface, chaotic microplasma discharges appear on the component surface, producing a glowing halo. These microdischarges exert both plasma-chemical and thermal effects on the coating and the electrolyte. At the sites of the discharges, a coating forms from oxidized forms of the base metal and electrolyte components. By selecting appropriate oxidation regimes and electrolyte compositions, coatings with different thicknesses, porosities, and properties can be obtained [164,165,166,167,168,169,170,171].

PEO electrolytes can be categorized as follows [165,166,168]:Electrolytes that do not contain components forming insoluble oxides, such as sulfuric acid, phosphoric acid, or alkaline solutions. In these electrolytes, the coating grows into the metal through its oxidation.Electrolytes containing cations or anions that form insoluble oxides or hydrolysis products, including aluminate, silicate-alkaline solutions, and solutions containing soluble phosphates, bicarbonates, and molybdates. After thermolysis, these electrolyte components are incorporated into the coating within the discharge zones, contributing to an increase in the coating thickness.

PEO regimes vary according to the type of current (direct, alternating, or alternating superimposed on direct), polarity, changes in electrical parameters (galvanostatic, galvanodynamic, potentiostatic, potentiodynamic, constant or decreasing power), the nature of the discharge (spark, microarc, arc, electrophoretic arc), and the degree of automation (manual, semi-automatic, automatic). Voltages in the bath range from 600 to 1000 V, current densities up to 30 A/dm^2^, and specific power consumption reaches 11,000 ÷ 30,000 W/dm^2^. Special surface preparation is generally not required. In practice, PEO is most often performed in mildly alkaline electrolytes using pulsed or alternating current.

Microarc discharges occur between the oxide film surface and the electrolyte, heating the film to temperatures of 1000 ÷ 2000 °C. At these temperatures, water undergoes thermal decomposition, forming atomic and ionized oxygen. High-temperature phases, such as corundum (α–Al_2_O_3_), form in the coating, and electrolyte components decompose and react with the metal oxides. Consequently, the PEO coating is not purely oxide but has a complex composition and structure. Approximately 70% of the oxide layer grows into the base metal, while only 30% extends beyond the original component dimensions. The metal/oxide/discharge/electrolyte system exhibits ionic conductivity, with current flowing through discharge channels, making porosity a necessary feature of the coating.

PEO coatings exhibit a layered structure:Outer (technological) layer—loose; in alkaline electrolytes with liquid glass, it consists of mullite (Al_2_O_3_ × 2SiO_2_).Inner (working) layer—dense, with high microhardness, composed of aluminum oxide (Al_2_O_3_).Transition layer—thin (0.01 ÷ 0.1 µm), located between the substrate and the oxide layer [168,169,174].

For titanium, PEO is carried out in electrolytes containing magnesium, calcium, or aluminum salts. The process involves the formation of a titanium dioxide (TiO_2_) layer at the surface under applied positive potential. When the voltage reaches a critical value, dielectric breakdown occurs, generating microarcs. These discharges locally heat the electrolyte in the pores, leading to the decomposition and formation of atomic oxygen, which diffuses into the metal surface and promotes oxide layer growth, reaching thicknesses up to 50 µm. Bath voltages may reach 200 V. The resulting coating is porous. The process involves sequential dielectric layer growth, breakdown, regrowth, and so on. Coating quality (hardness, porosity, breakdown voltage) depends on discharge intensity, duration of purely electrochemical and microarc phases, and the current magnitude affecting local temperature [164,166,168,169,172].

For magnesium and its alloys, PEO involves anodic oxidation through spark formation in aqueous electrolytes at voltages up to 700 V. Voltage, processing time, and other parameters are adjusted to achieve the desired coating thickness. High-voltage pulsed alternating current initiates plasma discharges on the magnesium surface when the dielectric breakdown threshold is exceeded. Coatings are obtained from alkaline electrolytes with silicate or phosphate additives (5 ÷ 50 g/L). Higher additive concentrations accelerate layer thickening. Proper voltage and current density are necessary to balance hardness and surface roughness. Typical voltage ranges from 200 to 700 V, with current densities from 0.1 to 15 A/dm^2^. Coating thicknesses on magnesium generally reach up to 150 µm, exhibiting excellent wear resistance, hardness, and protective properties without altering part dimensions, ensuring stable performance under high loads [165,171,173].

Typical PEO coating colors on magnesium include matte white, light gray, or off-white, depending on oxide composition and layer thickness. Color can be modified by adding pigments or adjusting pulsed AC parameters, or by immersion in a dye solution. Nanoporous PEO structures absorb dyes, producing durable decorative finishes, though dyeing times may be longer than for untreated magnesium [165,171,173].

PEO coatings can reach thicknesses of 300 ÷ 400 µm, with the ceramic-like composition determining their properties. Coatings can be produced in brown, black, blue, and white shades depending on the substrate alloy (e.g., D16—black/brown, V95—pink, AMg—beige, AK12—gray, titanium alloys—blue). Alloying elements contribute to coloration. Microhardness may reach 2500 kg/mm^2^, and wear resistance is comparable to tungsten carbide. PEO coatings exhibit chemical stability in atmospheric, acidic, and alkaline environments. Breakdown voltage averages 600 V and may rise to 2500 V with pore filling. Hardness and corrosion resistance depend on thickness, pore type, and size. Porosity ranges from 5% to 50%, with pore diameters from 0.01 to 10 µm. For coatings thicker than 5 ÷ 10 µm, the pore structure is complex, with branching and closed voids; porosity can be reduced to 2–3% by impregnation with polymers (PTFE), dyes, or oils [164,165,166,167,168].

Coating thickness is chosen according to function and operating conditions: 5 ÷ 10 µm for underlayers prior to painting, 20 ÷ 40 µm for decorative and corrosion protection, and 50 ÷ 100 µm or more for electrical insulation or high wear resistance. PEO is most commonly applied for thick coatings, as thin coatings are more economically produced by anodizing. The method allows for the formation of hard, electrically insulating coatings even on complex-shaped components [165,166,168].

Advantages of PEO include: no special surface preparation (degreasing, etching, brightening) since electric discharge self-cleans the surface; environmental friendliness due to absence of preparatory wastewater; ability to produce thick coatings (up to 400 µm) without electrolyte cooling; high wear resistance; and adjustable porosity, which can be reduced to 2 ÷ 3% through post-treatment. Limitations include the requirement for specialized high-voltage, high-current power supplies, increased energy consumption, making the method less suitable for large aluminum components, and significant surface roughening during coating formation [166,168].

## 3. Overlayers on Functional Coatings to Enhance Mechanical and Corrosion Properties

Adhesion and porosity are the key properties of protective and functional coatings, directly influencing their durability and performance. Adhesion determines the magnitude of mechanical stress that a coating can withstand before delamination, while porosity defines the permeability of coatings to corrosive environments. Porosity promotes the loss of adhesion between the substrate and the coating, which can serve as the starting point of coating degradation, especially in the case of anticorrosive coatings.

Porosity affects material properties such as strength, water absorption, thermal conductivity, and frost resistance. The lower the porosity, the higher the strength, frost resistance, and thermal conductivity, but the lower the water permeability. Based on the information presented in Section 2, it is evident that all the listed functional sublayers can be used as standalone coatings. However, to ensure reliable anticorrosive and physicomechanical properties, these coatings require additional sealing or enhancement of their mechanical performance, which improves their further independent operation.

When the listed coatings are used as functional sublayers, the presence of porosity may cause an anchoring effect, which can ultimately lead to improved protective, adhesive, and strength characteristics of the main coating. Additional sealing of functional sublayers can be achieved in various ways, such as using paint and varnish materials in the form of impregnations, varnishes, and paints; applying diffusion saturation of the coating material; or depositing an additional (in particular, duplex) coating by various spraying/immersion methods, etc.

In the case of duplex coatings, a wide range of combinations should be considered, not only zinc–polymer systems applied to metallic surfaces [174]. Depending on the application area of the coatings, additional sealing can significantly increase their resistance to corrosion and mechanical stress, enhance hydrophobic and anti-icing properties, and thereby extend the service life of metal structures and components.

### 3.1. Sealing of Coatings with Organic Materials

To protect the steel surfaces of structural and decorative constructions and products from the effects of aggressive corrosive atmospheres, paint and varnish coatings based on organic and waterborne formulations are commonly used.

The protective performance of such coatings largely depends on the nature of the film-forming base and the presence of special additives that enhance the anticorrosion and adhesion properties of the coating [175,176,177,178].

In this context, in [179], polymer-like films were obtained on a pre-oxidized steel surface from aqueous inhibitor compositions of vinyltrimethoxysilane (VS) combined with 1,2,3-benzotriazole (BTA), citric acid (CA), or 1-hydroxyethylidene-1,1-diphosphonic acid (HEDP). To further seal the film, a paint coating based on a water-dispersed styrene–acrylic resin (Lacryl 9930) was applied.

The structure, morphology, thickness, and structural properties of the polymer-like films were examined using X-ray photoelectron spectroscopy (XPS) and scanning electron microscopy (SEM). Corrosion and electrochemical studies were also carried out to evaluate the protective performance of the coatings.

It was established that the introduction of HEDP into the inhibitor composition led to its copolymerization with organosilane, resulting in the formation of a multilayer polymer-like film on the metal surface. The innermost layer of this film—similar to that formed without HEDP—is a nanoscale siloxane–azole layer. The intermediate siloxane–phosphonate layer forms through copolymerization reactions [180]. The outer siloxane–silanol layer, a product of silanol polycondensation, interacts with the components of the paint coating, ensuring the formation of a strong adhesive bond (Figure 5a).

Corrosion tests showed that after exposing the samples for 96 h in a salt spray chamber, significant corrosion damage was observed on their surfaces. Among all of the samples, the inhibitor composition VS+BTA+HEDP applied to the pre-formed OCC demonstrated the best performance (Figure 5b). The presence of the OCC on steel promoted the formation of barrier layers of iron oxides approximately 1 µm thick, as well as the development of a surface with a high number of active sites containing −OH groups, which facilitated the chemisorption of organosilane [181].

SEM analysis revealed that the OCC layer was loose and defective. Further modification of the OCC led to an increase in the thickness and a reduction in the porosity of the main coating, which was confirmed by corrosion testing and polarization resistance measurements. When a waterborne dispersion was applied onto the OCC, the total coating thickness increased to 36 µm. However, micrographs showed areas where Lacryl 9930 had delaminated from the OCC. Introducing the inhibitor composition VS+BTA+HEDP as an intermediate layer in the OCC/Lacryl 9930 system increased the total coating thickness to 45 µm and significantly reduced the corroded surface area due to the crosslinking of VS+BTA+HEDP with the OCC and the dispersion components.

In a similar study [182], researchers applied a polymeric conversion coating using styrene-acrylic dispersion (SAD) to metallic substrates made of low-carbon steel St3 and magnesium alloy MA2. The coating consisted of 70 vol% styrene–acrylic dispersion Lacroten 244 and 30 vol% ethylene glycol (EG), with 10 mMol/L of VS additives. This conversion coating was used as a primer for the subsequent application of an alkyd paint, Formula Q8. The formation mechanism of the protective coatings, as well as their adhesion properties, were investigated using XPS and SEM. The protective performance of the coatings was evaluated via electrochemical impedance spectroscopy after a two-week exposure of the samples in 3% NaCl at 60 °C. The adhesion tests were conducted in accordance with the ASTM D 4541 standard, which allows for the quantification of the strength of a coating’s adhesion to a metal surface.

Based on the EIS analysis data, it is possible to identify common trends for both metals before and after corrosion studies. Prior to testing on both metals, conversion coatings filled with Formula Q8 + SAD showed a wide range of phase angle changes characteristic of an intact coating. After 14 days of exposure to 3% NaCl at 60 °C, there was a slight change in the shape of the diagrams and a slight decrease in the phase angle in the frequency range 10^2^ ÷ 10^3^ Hz. Additionally, coating detachment sites were observed. However, there were some distinctive features in the corrosion behavior of the coatings. In the case of MA2, there was a sharp decrease in phase angle at high frequencies and a significant change in Bode diagram appearance (a characteristic double hump appearing with a minimum at 10 ÷ 100 Hz indicating coating detachment). Degradation of protective properties was less pronounced for St3. However, the formation of microcracks and pores, as well as the partial filling of these pores with corrosion products, has been observed. Adhesive studies have shown that applying Formula Q8 to the SAD primer conversion sublayer on metal substrates results in an increase in adhesive strength. For MA2, the adhesive strength was 4.6 MPa, and for St3, it was 5.1 Mpa, which was 5 to 6 times higher than the strength of coated substrates treated with Formula Q8 alone.

It was found that when MA2 interacts with the dispersion, the valence bonds of MgR_2_ become saturated, forming unsaturated [MgOC]ₙ^−^ compounds. The saturated organic compounds form conglomerates within the polymer coating lattice, where one terminal group of atoms is chemically bonded to the polymer coating, while the other part of the molecules forms regions of free chemical interaction with other molecules. As a result of relaxation processes, these molecular conglomerates cluster in such a way that voids are formed inside the polymer coating, with MgR molecules located along their periphery (Figure 6a).

In the case of steel, at the St3/conversion coating interface, chemisorbed poorly soluble iron–acrylic acid compounds are formed, strongly bonded to the steel substrate. Further chemical bonding of SAD with the alkyd paint occurs via polycondensation reactions between OH^−^ and COOH^−^ groups, forming surface layers of the metallic and alkyd enamels. The thickness of the near-surface conversion primer was 0.1 ÷ 0.3 µm, while the resulting upper layers formed a homogeneous polymer coating 1 ÷ 1.5 µm thick (Figure 6b). The presence of EG and VS promoted the formation of chemisorbed compounds on steel, whereas for MA2, these substances were incorporated into the polymer coating as uniformly distributed organometallic compounds.

In study [183], polymer–siloxane coatings were obtained on OCCs formed on the surfaces of magnesium alloy MA20 and aluminum alloy A5154 via vapor-phase deposition using VS and EG. During deposition, the concentrations of components in the vapor phase were controlled, the density of azeotropic mixtures was measured, and both temperature and optical parameters were monitored. This enabled the development of effective compositions of azeotropic mixtures and the balancing of partial pressures of the main components in the vapor–gas mixture. The researchers also carried out the vapor phase deposition of the coatings in a special chamber equipped with a disk heater for the evaporation of azeotropic mixtures: 80 vol% VS + 20 vol% butanol, 70 vol% EG + 30 vol% H2O, 60 vol% EG + 40 vol% phenol-methane.

After vapor-phase deposition, XPS and SEM were used to study the chemical composition and morphology of the resulting coatings. The protective properties of the coatings were evaluated using polarization methods, electrochemical impedance spectroscopy, and corrosion tests. SEM analysis confirmed the effectiveness of the vapor-phase deposition method. It was found that the structure of the coatings depends on the type of metal and the duration of treatment in the deposition chamber. For instance, on magnesium and aluminum without pre-oxidation and with 30 min of chamber treatment, the polymer–siloxane coatings were distributed unevenly across the metal surface, with exposed metal areas observed. When the samples had a pre-formed OCC and were treated for 90 min, the coatings were distributed evenly across the metal surface. However, on magnesium, more porous structures with a highly developed surface were formed, whereas on aluminum, a denser coating was produced. XPS analysis showed that ethylene glycol is an effective promoter of polymerization. Alongside the polymerization of VS silane, two- and three-dimensional structures formed, including polymolecular magnesium glycolate compounds. Thus, the mechanism of coating formation includes the following steps: evaporation of components from azeotropic mixtures; polymerization of VS silane with the participation of ethylene glycol; hydrolysis of siloxane groups; formation of a polymer structure. Polymerization of organosilane from the vapor–gas phase on the aluminum surface occurs differently: the polymer coating grows on the passive surface without the participation of aluminum ions, resulting in a thinner and denser coating (Figure 7).

Electrochemical and corrosion tests demonstrated that when an OCC is present on the aluminum and magnesium surfaces, the deposited polymer–siloxane coatings exhibited higher protective properties compared to both the bare metal surfaces and polymer–siloxane coatings formed on non-oxidized aluminum (ΔE = 465 mV) and magnesium (ΔE = 633 mV). Since polymer–siloxane coatings form chemical bonds with the oxidized layers of aluminum and magnesium alloys, this effect can be exploited in the development of primers prior to coating application. The study demonstrates the potential of the developed technology for protecting aluminum and magnesium alloys from corrosion, especially under aggressive environmental conditions.

The development of effective methods for forming protective layers on aluminum alloys, which are widely used in the aerospace, automotive, and construction industries, is of particular importance. For instance, in [79], the influence of various methods for forming protective coatings on the corrosion resistance of the aluminum alloy EN AW 6060 was investigated. In this study, aluminum samples with the following oxide-conversion layers were examined: clear anodizing, black anodizing, and a combined coating (anodizing + application of a paint coating). The paint coating was applied using electrostatic spraying of an architectural polyester powder paint in accordance with the Qualicoat standard. Corrosion behavior of the samples was studied using potentiodynamic and EIS measurements in 3.5% NaCl at room temperature. SEM analysis showed that all coatings had a hexagonal columnar structure with nanopores, and the coatings were distributed uniformly across the aluminum substrate.

For clear anodizing, the OCC thickness was 17 µm, while for black anodizing, it was 18 µm. For the combined OCC, the anodized layer thickness ranged from 5 to 8 µm, with an additional paint layer also 5 ÷ 8 µm thick. Potentiodynamic measurements showed that for the bare EN AW 6060 sample, the corrosion potential (E_corr_), corrosion current density (I_corr_), and corrosion rate (V_corr_) were –0.756 mV, 9 × 10^−7^ A/cm^2^, and 0.03 mmpy, respectively. For the clear anodized sample, the corrosion rate decreased 30-fold compared to the bare alloy, with V_corr_ = 0.0009 mmpy, E_corr_ = –0.702 mV, and I_corr_ = 2.72 × 10^−8^ A/cm^2^. For black anodizing, the corresponding values were −0.722 mV, 4 × 10^−8^ A/cm^2^, and 0.0014 mmpy. The highest efficiency was observed for the sample with the combined OCC: E_corr_ = −0.722 mV, I_corr_ = 9 × 10^−10^ A/cm^2^, and V_corr_ = 0.00003 mmpy, which was 1000 times lower than the bare EN AW 6060 and 30 times lower than the clear anodized sample. EIS measurements via the charge transfer resistance (R_ct_) parameter correlated with the corrosion current density data, i.e., higher R_ct_ corresponded to lower I_corr_: EN AW 6060 R_ct_ = 13 kΩ, clear anodizing R_ct_ = 1.48 MΩ, black anodizing R_ct_ = 0.74 MΩ, and combined coating (considering multilayer structure) R_ct_ = 0.47 MΩ. Thus, the use of combined OCC improved the mechanical properties and adhesion of the coating to the substrate, demonstrating reliable protection. The study indicates the applicability of such coatings in aggressive environments.

### 3.2. Sealing of Functional Sublayers via Additional Coatings

Considering the wide use of black anodic coatings in the aerospace industry for the passive thermal control of satellites and absorption of scattered light in optical components, it is necessary for these coatings to provide the high corrosion and wear resistance of aluminum alloys. Studying the influence of various coloring methods on the characteristics of porous anodized layers is therefore particularly important. In the cited work, the electrochemical properties of porous layers of different types of black anodic coatings (standard sulfuric acid anodizing (SAA), black dyeing (BD), inorganic coloring (IC), and electrolytic coloring (EC)) on the aluminum alloy Al6061-T6 were studied, followed by an evaluation of their corrosion resistance [184].

The black anodizing process included the following main stages: surface pre-treatment, anodizing, coloring (black anodizing), and sealing. Additional sealing was performed by immersing the samples in boiling demineralized water for 30 min after each black anodizing process. The pore-filling mechanism included the dissolution of anhydrous aluminum oxide from the pore walls, the formation of amorphous layers at the pore openings, and pore filling with hydrated aluminum oxide. As a result of sealing, a specific structure was formed: the outer layer was crystalline, the intermediate layer was hydrated oxide, the inner layer consisted of partially filled pores, and the barrier layer was compact.

After sealing, the pore-filling quality was evaluated using electrochemical impedance spectroscopy, linear polarization, and scanning electron microscopy. Quality control was based on parameters such as the resistance of the porous structure (R_p_), capacitance characteristics, stability of the barrier layer, and corrosion resistance. SEM analysis showed that after prolonged exposure to an aggressive environment (15 days in 3.5% NaCl), none of the coatings showed signs of general or localized corrosion. EIS measurements recorded that the impedance spectra of samples with black coloring (BD, IC, EC) demonstrated the characteristic three regions for properly sealed anodic oxide layers. EC coloring produced a fundamentally different layer structure compared to other methods: a thin metallic tin layer was deposited, a specific pore structure formed, and surface leveling was observed. It was determined that the porous structure resistance (R_p_) in BD-colored samples was lower than in other coating types. The damage coefficient (D) after 360 h of immersion in 3.5 wt% NaCl was minimal for samples with traditional sulfuric acid anodizing (SAA). All tested coating samples demonstrated high corrosion resistance under prolonged exposure to aggressive environments. These results effectively characterize the performance of various types of black anodic coatings in corrosive conditions, supporting their practical applicability.

Another approach to additional sealing is the immersion of coated samples in modifying solutions, such as corrosion inhibitors or hydrophobizing agents [124,185]. In particular, in [124], aluminum alloy AMG-3 with porous OCC formed in solutions containing permanganate (KMnO_4_, NiSO_4_), and molybdate ions (Mo, NH_4_NO_3_; Mo, NaF) underwent pore-filling by 30 min exposure to a hot solution (96 ÷ 100 °C) of a corrosion inhibitor. The protective performance of the coatings was evaluated via anodic polarization curves from AMG-3 samples with and without filled OCC. Tests were performed at room temperature in a borate buffer (BB) solution at pH 7.4 with 0.01 M NaCl. As a reference for evaluating the quality and protective properties of the resulting OCC, samples treated with a chromate oxidation solution (5.5 g/L NH_4_F, 0.9% CrO_3_, 7% H_3_PO_4_) at 40 °C for 10 min were used (Figure 8).

As can be seen in Figure 8a, the best protective performance was demonstrated by the coating obtained from the chromate solution. In this case, the potential shifted by 0.6 V in the positive direction. In contrast, the OCCs that formed from both molybdate solutions actually accelerated the onset of corrosion relative to the uncoated AMG-3 sample. This effect is attributed to the presence of through-porosity in the OCCs, which allows Cl^−^ ions to penetrate and accelerate metal degradation. The OCC from the permanganate solution shifted the potential by more than 0.2 V in the positive direction relative to the reference sample, significantly outperforming the molybdate-based coatings. This indicates the continuity of the resulting coating, although its protective properties are still inferior to the chromate conversion coating.

From Figure 8b, it is evident that additional sealing in a corrosion inhibitor solution significantly improved the protective properties of coatings obtained from molybdate solutions. This confirms the previous conclusion regarding their porosity. The pores were filled with adsorbed inhibitor, which prevented Cl^−^ ions from reaching the metal surface. The molybdate solution activated with NH_4_NO_3_ produced a filled coating whose quality matched that of the chromate solution coating. The molybdate solution activated with NaF allowed for a positive shift of 0.15 V relative to the pitting corrosion potential of the chromate-coated sample. However, the potential shift for the permanganate-based coatings did not exceed 0.1 V, indicating low porosity and ineffective filling.

Thus, additional sealing of OCCs by immersion in a corrosion inhibitor solution can enhance their protective properties up to the level of standard chromate coatings.

The magnesium alloy WE43 is widely used in the aerospace industry, in transmission manufacturing, and for biomedical implants due to its excellent mechanical properties at elevated temperatures. However, its low electrochemical potential leads to poor corrosion resistance, which hinders the formation of durable protective layers. Therefore, there is a need to develop effective corrosion protection methods to ensure the reliable safeguarding of magnesium alloy products.

As shown in [108], anodizing alone does not always provide sufficient protection against corrosion damage. In this context, the authors applied additional sealing of anodized OCCs on the WE43 magnesium alloy through modification with organic compounds (folic acid (FA) and lauric acid (LA)) to enhance corrosion resistance and protective properties. The additional sealing of anodized OCCs was performed by immersing the samples in aqueous solutions of FA and LA for 10 ÷ 15 min, followed by drying at 40 °C.

Scanning electron microscopy revealed that the structure of anodized OCCs after modification with FA or LA became denser compared to unmodified anodized OCCs, which contained pores and cracks. Gaps were observed between the anodized OCCs and the magnesium substrate. In the FA-modified anodized OCC, the gaps were reduced, whereas in the LA-modified OCC, the gaps were almost entirely eliminated.

Potentiodynamic studies in a 3.5 wt% NaCl solution showed that the current density (I_corr_) for the magnesium substrate was 47.841 µA/cm^2^, 3.689 µA/cm^2^ for the anodized conversion coating (CC), and for the CC modified with FA and LA, the I_corr_ values were 0.399 and 0.018 µA/cm^2^, respectively. It is evident that the LA modification of the anodized CC was the most effective, as confirmed by EIS analysis: the impedance values for the LA-modified coating improved by three orders of magnitude (6.36 × 10^6^ Ω × cm^2^) compared to the magnesium substrate (1.48 × 10^3^ Ω·× cm^2^). Moreover, it was observed that the initially hydrophilic anodized CC became hydrophobic after LA modification. In this case, the contact angle was 130.3°, whereas for FA modification it was 14.3° (hydrophilic), i.e., smaller than that of the magnesium substrate (35.6°) and the unmodified anodized CC (15.7°). Thus, it can be concluded that LA modification of the anodized CC resulted in the formation of a hydrophobic surface, which prevented corrosion damage to the coating. Combined SEM and electron probe microanalysis revealed that both FA and LA penetrated the pores and cracks of the anodized CC (Figure 9).

In the case of LA modification, defects were gradually eliminated, and gaps decreased until they disappeared. These results demonstrate the high effectiveness of the proposed method for sealing anodized CCs and highlight its potential for practical application in protecting magnesium alloys from corrosion.

The development of technologies for producing amorphous plasma coatings, which combine high strength with corrosion resistance, is of particular importance. At the same time, the mechanisms of corrosion behavior in such systems and methods for enhancing their protective properties are insufficiently studied [186,187,188,189]. In work [185], the main objective was to investigate the corrosion-electrochemical behavior of metal (substrate)/plasma-sprayed coating systems and find ways to mitigate the effects of porosity without altering the structural state of the coatings. As coating materials, gas-atomized powders of two alloys with compositions close to eutectic were used: FBH6-2 and SHS7574HV1. Low-carbon steel St3 served as the substrate material. The thickness of the sprayed FBH6-2 coatings was 60, 100, and 210 ± 10 µm, while for SHS7574HV1, it was 60 and 100 ± 10 µm. For additional sealing, the coatings were treated with the hydrophobizing agent Anacrol-2051 (active component—urethane dimethacrylate), which possesses high penetration ability and is recommended for sealing micropores and microcracks on metal surfaces. Coating impregnation was performed by immersing the samples in the solution at room temperature (~25 °C) for 2 min, followed by thermal treatment for 2 h at 80 °C to accelerate curing of the sealant. Structural characteristics of the coatings were analyzed using X-ray phase analysis. The composition and element distribution profile through the coating thickness were determined by SEM analysis. Corrosion-electrochemical behavior was studied using chronopotentiometry and potentiodynamic voltammetry in 3% NaCl. It was found that the FBH6-2 coating contained a predominantly amorphous phase (96%) with a small amount of α–phase, whereas the SHS7574HV1 coating exhibited a fully amorphous structure. The primary factor affecting corrosion degradation was identified as the presence of through porosity. Increasing the coating thickness was shown to reduce the influence of porosity. For FBH6-2, at a thickness of approximately 200 µm, the pores were almost completely filled with corrosion products from the coating itself, blocking electrolyte access to the substrate metal. For SHS7574HV1, a similar effect was achieved at a coating thickness of about 100 µm. It was determined that SHS7574HV1 exhibited higher corrosion resistance compared to FBH6-2 due to the presence of molybdenum in its composition. Molybdenum contributes to self-passivation and enhances the coating’s corrosion resistance. It was demonstrated that impregnation with the hydrophobizing agent Anacrol-2051 effectively seals pores, preventing access of the corrosive environment and thereby protecting the substrate metal. These results have significant practical implications for the development of durable protective coatings for metal structures operating in aggressive environments.

Literature data indicate that surface modification of corrosion-prone light alloys to impart hydrophobic properties significantly enhances their corrosion protection. The high adhesion of PEO coatings to the metallic substrate, their good anticorrosion performance, and developed surface morphology make these coatings a promising target for further modification, including the creation of composite coatings using hydrophobic agents and nanoparticles. The formation of hydrophobic (HP) and superhydrophobic (SHP) coatings, as well as the study of their electrochemical behavior, represent important steps in developing corrosion protection for materials that is effective not only under atmospheric conditions, but also in aggressive environments.

For example, in works [190,191], the formation of composite coatings on aluminum alloy AMg3 using the PEO method followed by the application of a hydrophobic layer was studied to reduce the wettability of the resulting coatings by corrosive media. Additional sealing of the PEO coatings involved treatment with a hydrophobic agent solution in the form of methoxypentadecafluorooctyloxypropylsilane, as well as a dispersion of nanoscale silicon dioxide particles in a superhydrophobic agent (SHP). To evaluate the influence of deposition conditions for the hydrophobic layer on the PEO coating, the samples were divided into two groups. Samples in the first group received no additional treatment, while in the second group, before the formation of the superhydrophobic layer, the samples were treated in boiled bidistilled water for 60 min. This pretreatment increased the number of chemisorption-active sites on the surface, i.e., the concentration of –OH groups, and sealed the microtubes by forming a dense surface layer of Al(OH)_3_.

X-ray phase analysis of the initial PEO coating showed that it contained only γ–Al_2_O_3_. Cross-sectional data of the first group of samples clearly illustrate that the resulting SHF layers replicated the macroscopic relief of the PEO coating, with significant differences appearing only at the micro- and nanoscales. PEO coatings consisted of microtubes formed on the AMg3 aluminum alloy substrate. This structure is typical for layers formed in tartrate-containing electrolytes during the early stages of oxidation. The self-organization process of nanoparticles in a dispersion containing nanosized SiO_2_ particles, a hydrophobic agent, and a nonvolatile dehydrated dispersing medium led to the deposition and aggregation of SiO_2_ nanoparticles, forming multimodal roughness (Figure 10). EDX results showed a higher silicon content in the SHP coating, supporting the assumption regarding the origin and nature of these aggregates.

SEM images of HP and SHP layers from the second group of samples demonstrated that their surfaces differed significantly from the original PEO layer, exhibiting a more multimodal relief but lacking a uniform surface structure. At the same time, since the hydrophobic agent deposits as a monolayer on the sample surface, the observed differences in SHP morphology between the first group and the second group (Figure 10) can be attributed to the prior boiling pretreatment. The structure of the initial PEO coating promoted the formation of an interface with trapped air bubbles adhered to surface irregularities when in contact with an aqueous medium. At the same time, one cannot exclude the displacement of the trapped air during prolonged contact of the surface with the liquid phase and its penetration into the microtubes. In this case, the protective performance of the HP and SHP coatings will largely depend on the barrier properties of the nonporous PEO layer (Figure 10).

However, a comparison of the morphology of the original samples and the boiled samples suggests that the open microtubes containing the deposited HP agent and SiO_2_ nanoparticles can retain air bubbles much longer than microtubes sealed by an impermeable aluminum hydroxide layer formed during boiling. On the surfaces of the HP and SHP samples obtained on the PEO coatings after boiling, inhomogeneities were observed, which increased the contact area with the corrosive solution, and consequently reduced the resistance compared to the non-boiled samples.

This explains why, under potentiodynamic polarization, the HP and SHP coatings of the first group demonstrated higher stability compared to coatings of the second group. The corrosion current density values calculated from the polarization curves for the SHP-coated sample were 4.6 × 10^−14^ A/cm^2^, which were more than seven orders of magnitude lower than for the uncoated sample (6.1 × 10^−7^ A/cm^2^), two orders of magnitude lower than for the HP coating (4.8 × 10^−12^ A/cm^2^), and nearly four orders of magnitude lower than for the base PEO coating (3.5 × 10^−10^ A/cm^2^).

For the sample with the surface layer after boiling, the corrosion current density decreased more than 1.5 times (j_k = 1.9 × 10^−10^ A/cm^2^) compared to the original PEO coating (j = 3.5 × 10^−10^ A/cm^2^), and more than four orders of magnitude compared to the unprotected aluminum alloy (j = 6.7 × 10^−7^ A/cm^2^). This indicates that the microtubes were effectively sealed by the forming aluminum hydroxide, resulting in improved corrosion resistance of the PEO coating.

Wettability studies using the sessile drop method in 3% NaCl with coatings from the first group of samples (without boiling in water) showed that the original PEO layers were hydrophilic, with a contact angle of (46 ± 3)°. Samples with HP coatings on PEO layers without prior boiling exhibited a contact angle of (160 ± 7)°. However, the rolling angle for the HP coating was 18 ± 9°, indicating a heterogeneous multimodal roughness of this coating. Surface regions with less pronounced hierarchical roughness act as anchors, holding the drop in place when the sample is tilted. The superhydrophobicity of coatings obtained by applying the HP agent with SiO_2_ nanoparticles was indicated by a higher contact angle (165 ± 3°) and a more than twofold decrease in the rolling angle (8 ± 3°) compared to the HP coatings (18 ± 9°).

Analysis of the wettability of the second group of PEO-coated samples showed that boiling reduced the contact angle from (46 ± 3)° to (36 ± 3)°. After treatment with UV radiation in an ozone environment and application of the HP agent, the increased number of hydroxyl groups resulting from boiling the PEO coatings promoted denser deposition of the HP agent, which in turn increased the contact angle to (162 ± 2°) and reduced the rolling angle to (14 ± 2)°. Deposition of the HP agent with SiO_2_ nanoparticles further increased the contact angle (166 ± 3°) and decreased the rolling angle (12 ± 3°) due to the formation of a multimodal surface relief. Nevertheless, the rolling angle remained above 10°. Likely, on the SHP layer, the rolling droplet slows down when encountering regions with heterogeneous roughness, which increases the tilt angle of the sample required for the drop to roll off.

An acrylic polymer applied to the surface of PEO coatings on ML5 and MA2 magnesium alloys, obtained in an alkaline phosphate-aluminate electrolyte, significantly enhances their corrosion resistance. In study [170], the effect of polyanion electrolysis n[Si_x_O_y_]_m_^−^ and the acrylic polymer on the corrosion resistance of coatings formed on PEO-treated ML5 and MA2 alloys was investigated. The polymer coating was applied to the PEO coatings by immersing the samples for 10 min in a solution containing the acrylic polymer [192]. Afterwards, the samples were air-dried at room temperature for 1 h.

It was experimentally established that small additions (0.5 ÷ 2 g/L) of NH_4_F to aqueous solutions containing high concentrations (100 ÷ 200 g/L) of technical liquid glass (TLG) enable the formation of thick (>60 µm), almost uniform PEO coatings on different areas of the sample surfaces. In particular, an aqueous solution containing 150 g/L TLG and 1 g/L NH4F was found to be optimal compared to other alkaline-silicate electrolytes, as PEO in this solution provided a high coating growth rate (approximately 2.8 µm/min) at a current density of 4 A/dm^2^, with minimal surface roughness. In addition, the coatings obtained in this solution exhibited satisfactory adhesion to the metallic substrate.

On the ML5 alloy, coatings with thicknesses of approximately 30 and 60 µm detached at 15 ± 4 and 22 ± 2 MPa, respectively, leaving inner layers with thicknesses of 4 ± 2 and 15 ± 5 µm. On the MA2 alloy, coatings around 30 µm thick detached from the metal substrate at 12 ± 4 MPa, while coatings approximately 60 µm thick had their outer layers detached at 21 ± 3 MPa, leaving inner layers with thicknesses of 3 ± 2.5 and 25 ± 6 µm. The remaining inner layers exhibited an adhesion greater than that of VK-27 adhesive.

The corrosion resistance of the coatings was significantly improved after the application of the acrylic polymer (PEO + AP). Corrosion tests showed that during the exposure of the magnesium alloys with PEO + AP coatings for 300 h in 3% NaCl, no signs of localized corrosion or hydrogen evolution were observed. In contrast, for the ML5 alloy with coating thicknesses of 30 and 60 µm, the incubation period was 38 and 42 ± 3 h, respectively. For the ML2 alloy with coating thicknesses of 30 and 60 µm, the time to the appearance of the first corrosion defect was 18 and 33 ± 2 h, respectively. Electrochemical tests of the magnesium alloy samples with PEO + AP coatings confirmed the results of the corrosion studies (Figure 11).

The corrosion potential values for the ML5 and MA2 alloys shifted toward the positive region, while the corrosion currents decreased significantly. The greatest reduction in corrosion current was observed due to the applied acrylic polymer on coatings with intermediate thicknesses of 30 µm on the ML5 alloy and 60 µm on the MA2 alloy—the corrosion currents of the alloys decreased approximately by factors of 4.2 × 10^4^ and 5 × 10^5^, respectively. Thanks to the acrylic polymer, the corrosion resistance of coatings with thicknesses around 30 µm on the ML5 alloy and around 60 µm on the MA2 alloy increased by factors of 13 and 4.1 × 10^2^, respectively.

It should be noted that using the acrylic polymer without a PEO-prepared coating resulted in blistering at various areas of the samples and intensive hydrogen evolution, leading to pitting even during relatively short exposure in 3% NaCl solution. The incorporation of the acrylic polymer into the outer layer of the coatings ensures its adhesion to them. Therefore, PEO + AP coatings enable a multiple-order increase in the corrosion resistance of the MA2 and ML5 alloys. Thus, PEO + AP coatings can be recommended for the long-term corrosion protection of various magnesium alloy components in chloride-containing environments.

Another approach to the surface modification of coatings was considered in [193]. A bilayer W/WC_1−x_ coating applied on a copper substrate was subjected to laser texturing, followed by treatment in a solution of octadecylphosphonic acid (ODPA). It was found that laser treatment leads to decomposition of the initial tungsten carbide, resulting in a multiphase coating composed of larger crystalline forms (as evidenced by the broadening of peaks in the initial state and after treatment, Figure 12a). At the same time, laser treatment at maximum power preserved a portion of the coating thickness in an undamaged state (Figure 12b), and after ODPA treatment, the surface became superhydrophobic (Figure 12c) with a contact angle of approximately 168°.

A similar approach can also be applied to layers of other materials, provided that methods for creating polymodal roughness and forming a superhydrophobic surface, as well as techniques for their deposition onto target substrates, are known. For instance, this approach may be relevant for coatings produced by thermal spraying [194], electroplating [195], and even for organic coatings [196].

As above-mentioned, another option for additional sealing is the use of duplex coatings. Duplex coatings generally refer to a combination of two or more coating layers applied to a surface to provide corrosion protection or decorative properties. At present, such coatings are widely used in various fields, including the automotive and aerospace industries, medicine, microelectronics, and jewelry manufacturing.

Die-casting molds made from ABS and PVC mixtures are highly susceptible to corrosion by HCl, which significantly reduces their service life [197,198]. To address this problem, a duplex coating applied to Q235 steel was investigated [199]. The studied coating consisted of the following layers: the inner layer was a thick corrosion-resistant NiCr–30Cr_3_C_2_ coating (containing an intermediate layer with 30 wt% Cr_3_C_2_). This layer was deposited on the steel substrate using high-velocity oxygen fuel (HVOF) spraying. The outer layer was a TiSiN nanocomposite coating deposited on top of the HVOF coating by physical vapor deposition (PVD).

Phase composition analysis showed that the inner layer of the duplex NiCr–30Cr_3_C_2_ coating consisted of γ–NiCr and Cr_3_C_2_ phases. This layer exhibited a high density of defects such as pores, cracks, and unmelted regions. The outer TiSiN layer of the duplex coating consisted of FCC nanocrystals embedded in an amorphous Si_3_N_4_ matrix.

Electrochemical corrosion tests of both single-layer HVOF and PVD coatings, as well as the duplex HVOF–PVD coating, were conducted in 10% HCl at 65 °C. It was found that the duplex HVOF–PVD coating demonstrated significantly higher corrosion resistance compared to the single-layer coatings. The low corrosion resistance of the HVOF coating (NiCr–30Cr_3_C_2_) was attributed to severe passive film breakdown caused by chloride ions and the presence of numerous defects (pores, cracks, etc.), which facilitated electrolyte penetration.

The poor corrosion resistance of the PVD coating (TiSiN) was due to delamination induced by combined pitting and interfacial (galvanic and chemical) corrosion, as well as cracking. Pitting corrosion developed as micro-particles grew, initiating defects that propagated toward the steel substrate once the corrosive medium entered them. The interaction of the electrolyte with steel involved a series of interfacial reactions—including displacement, dehydration, hydrolysis, and redox processes—accompanied by cracking. The combination of these effects led to accelerated degradation of the PVD coating.

The excellent corrosion resistance of the duplex HVOF–PVD coating resulted from the chemical inertness of the TiSiN layer and the presence of the corrosion-resistant NiCr–30Cr_3_C_2_ interlayer, which prevented the initiation of interfacial corrosion and crack formation. Thus, it was demonstrated that the duplex HVOF–PVD coating can be effectively used in die-casting molds operating in aggressive corrosive environments. Moreover, as shown in [200,201,202,203], such duplex HVOF–PVD coatings possess higher load-bearing capacity and wear resistance compared to single-layer coatings produced by either HVOF or PVD alone.

Thermal barrier coatings (TBCs) are widely used in modern industry, especially in the aerospace sector. As performance and reliability requirements for gas turbine engines continue to increase, improving the thermal stability and corrosion resistance of materials has become a particularly pressing challenge [204,205]. One effective approach to achieving high-performance thermal protection is through duplex coating systems.

The main objective of the study in [206] was to investigate and compare the protective performance of duplex coatings applied to Inconel 718 using high-velocity oxygen fuel (HVOF) spraying and electron beam physical vapor deposition (EB-PVD) under oxidative and corrosive environments. The inner (bond) layer of the duplex coating, approximately 100 µm thick, was deposited using the HVOF process and consisted of a CoNiCrAlY alloy. The porosity of this bond coat was 2.5 ± 0.5%.

The outer (ceramic) coating was deposited using the EB-PVD method and examined in two configurations: a single-layer YSZ (yttria-stabilized zirconia) coating about 100 µm thick, composed of ZrO_2_ with 8 wt% Y_2_O_3_, and a double-layer YSZ/Gd_2_Zr_2_O_7_ (gadolinium zirconate) system, with each layer approximately 100 µm thick. The layers showed a distinct contrast due to the high gadolinium content in the upper layer. No material deficiency was observed at the interface between the layers, and both exhibited integrity and strong adhesion. The porosity of the ceramic layers was measured at 1.6 ± 0.3%. The duplex coatings displayed a columnar microstructure that provided good thermal shock resistance.

Isothermal oxidation tests were carried out at 1000 °C for 8, 24, 50, and 100 h. During oxidation, oxygen penetrated through the pores of the top ceramic layer to the bond coat (CoNiCrAlY), where a dense thermally grown oxide (TGO) layer formed at the bond coat/ceramic interface. The TGO layer, composed primarily of Al_2_O_3_, grew according to a parabolic rate law. Its growth was uniform and maintained structural integrity, with no phase transformations detected in the primary coating components under the test conditions. The TGO acted as a diffusion barrier, slowing further oxidation of the bond coat.

Corrosion tests were performed in a molten salt mixture of 30% NaCl, 35% Na_2_SO_4_, and 35% V_2_O_5_ at 1000 °C for cycle durations of 5, 10, 15, and 20 h. These tests revealed several phase transformations, including the formation of monoclinic zirconia (m-ZrO_2_), YVO_4_, GdVO_4_, NaAlO_2_, and Na_2_CrO_4_. Microstructural changes were also observed, such as the development of elongated rod-like features, the filling of intercolumnar gaps, and an overall increase in coating density.

Under high-temperature corrosive exposure, the CoNiCrAlY/YSZ system, lacking an additional protective layer, exhibited active YVO_4_ formation in the upper layer, leading to accelerated corrosion. In contrast, the CoNiCrAlY/YSZ/Gd_2_Zr_2_O_7_ system benefited from the gadolinium zirconate layer acting as a sacrificial barrier—the top layer degraded first, preserving the integrity of the underlying YSZ layer.

Mechanical testing showed that the hardness of the YSZ layer increased from 662.8 HV before corrosion testing to 858.6 HV afterward. The filling of intercolumnar gaps with corrosion products and sintering effects at high temperatures between the TGO and YSZ layers reduced thermal insulation performance and strain tolerance in the upper layers. For the Gd_2_Zr_2_O_7_ layer, hardness values before and after corrosion testing were 458.9 HV.

Thus, the double-layer duplex system demonstrated superior corrosion resistance due to the sacrificial degradation of the gadolinium zirconate top layer, which protected the main YSZ coating.

Proton exchange membrane fuel cells (PEMFCs) are a promising energy source for transportation and portable electronic applications due to their high energy efficiency and environmental friendliness. However, for the broader commercial adoption of PEMFCs, several challenges must be addressed: enhancing corrosion resistance in acidic environments, reducing interfacial contact resistance between metallic bipolar plates (BPPs) and other cell components, and improving electrical and thermal conductivity as well as mechanical durability.

In this context, multilayer coatings composed of niobium (Nb) and tantalum (Ta) represent an innovative solution for protecting titanium-based BPPs. These coatings combine excellent corrosion resistance in acidic media with low interfacial contact resistance and offer a more cost-effective alternative to noble metals.

In the study by [207], single-layer (Nb and Ta) and multilayer (Ta/Nb and Nb/Ta) anticorrosion coatings were developed and investigated on titanium substrates using magnetron sputtering for potential application in PEMFC bipolar plates. Comprehensive characterization was carried out using advanced physical analysis techniques, electrochemical testing, and mechanical evaluations to assess performance under simulated operating conditions.

Morphological analyses using SEM and EDX revealed that the coatings possessed dense, continuous structures without visible cracks or delamination. Elemental mapping confirmed a uniform distribution of Nb and Ta across the surface, with no detectable impurities, pores, or interfacial defects between the layers. The crystallographic compatibility of the multilayer coatings with the titanium substrate (both exhibiting a body-centered cubic structure with a lattice parameter of 3.30 Å) contributed to improved interlayer adhesion.

The measured coating thicknesses were as follows (in nm): Nb—172 ÷ 344, Ta—224 ÷ 448, Ta/Nb—224/344, Nb/Ta—353/219. Electrochemical testing demonstrated that the multilayer Nb/Ta coating on titanium achieved the lowest corrosion current density, below 1 µA/cm^2^, indicating strong passivation behavior and enhanced corrosion resistance. Moreover, this coating exhibited a significant reduction in interfacial contact resistance compared with other configurations, even after 6 h of chronoamperometric testing, meeting the technical specifications set by the U.S. Department of Energy.

Contact angle measurements further revealed that the Nb/Ta coating increased the surface contact angle, indicating a transition in surface wettability from hydrophilic to hydrophobic.

Overall, the results demonstrate the high potential of multilayer Nb/Ta coatings as protective layers for titanium bipolar plates, offering excellent corrosion resistance, low contact resistance, and improved surface properties under PEMFC operating conditions.

In [208], a superhydrophobic duplex OCC was deposited on the surface of carbon steel by electrodeposition. The first OCC layer consisted of nickel, while the second was composed of trivalent chromium compounds (Ni/Cr-20—20 s of chromium deposition, Ni/Cr-120—120 s of chromium deposition). To impart hydrophobic properties, the duplex OCC was further sealed in a 0.01 M myristic acid solution. As a result, the developed duplex OCC exhibited high microhardness, reaching 754 ± 4 HV. The water contact angle reached 167.9 ± 2.4°, indicating the SHP properties of the duplex coating after additional treatment. The superhydrophobic properties were retained even after mechanical testing (100 abrasion cycles), i.e., the coating demonstrated high wear resistance.

The corrosion resistance of the coatings was evaluated using potentiodynamic polarization and EIS measurements in 3.5% NaCl at pH 5.9. The Ni/Cr-20 sample exhibited the best electrochemical performance, with E_corr_ = −0.073 V and I_corr_ = 2.525 × 10^−8^ A/cm^2^, respectively. Analysis of the polarization curves revealed lower cathodic slopes compared to anodic ones, effective surface protection in the cathodic region, and a significant reduction in current density variation under positive potential shift. The EIS data also correlated with the potentiodynamic results: the Ni/Cr-20 sample showed the largest capacitive arc, with broad and tall phase peaks in the mid-frequency region, indicating the absence of corrosive ion penetration. The impedance modulus for Ni/Cr-20 reached its maximum value.

This efficiency is attributed to the optimal structure of the outer layer formed within 20 s of chromium deposition. A flaky microstructure developed, preserving the micro/nano-relief of the nickel sublayer. Chromium was evenly distributed in the outer layer of the duplex OCC. Thus, the Ni/Cr-20 duplex OCC demonstrates an optimal combination of mechanical, chemical, and structural characteristics that provide maximum corrosion protection.

Despite the improvement of the protective properties of PEO layers through sealing via HP and SHP treatments, complete corrosion protection cannot be guaranteed, since over time, moisture may penetrate under the polymer layer, leading to a gradual degradation of anticorrosive performance [190,191]. In [113], it was shown that chemical-catalytic deposition in solutions containing NiP or CoP can provide an additional sealing of PEO coatings on magnesium and its alloys. Alternating deposition is possible—first in one bath to obtain a Ni–P alloy layer, then in another to form a Co–P outer layer. The sequence of layers may vary (Figure 13). The total thickness was 25 µm: 15 µm—PEO base layer, 10 µm—NiP or CoP alloy layer.

This sealing method is particularly important for critical components, as it ensures the complete elimination of through porosity. A uniform coating is formed over the entire surface: there is no morphological inheritance of the previous layer by the subsequent one, and the outer layer effectively covers any remaining through pores of the underlying coating.

Exposure tests in a high-humidity chamber demonstrated a 100% protective effect for the ML5 magnesium alloy. Complete elimination of through porosity was achieved, along with a reduction in coating deposition time.

This duplex coating formation method is suitable for the treatment of magnesium components with complex geometries (such as bushings, shafts, injectors, and pistons). The obtained results open up new prospects for the corrosion protection of magnesium alloys and can be applied across various industrial fields.

### 3.3. Top-Coating Approaches for Improving Mechanical Performance

Improving the performance of cutting tools is one of the key directions in modern metalworking. Existing single-layer coatings no longer meet contemporary wear-resistance requirements, which necessitates the development of new coating compositions and architectures.

In study [209], multilayer coatings of two types—TiAlN/TiAlMeN (where Me = Cr, Zr, Mo, Si) and TiZrN/TiZrMeN (where Me = Al, Cr, Mo, Si)—were deposited on the surface of cemented carbide inserts (grade MC146) via ion-plasma sputtering to enhance the cyclic crack resistance of cutting tools based on the evaluation of their physicomechanical properties.

It was found that the cyclic crack resistance of the multilayer coatings was 3.8 to 10 times higher than that of single-layer coatings. The maximum crack resistance was achieved for coatings with Cr, Si, and Al alloying in the top layer, with the optimal top-layer thickness being 40 ÷ 60% of the total coating thickness.

Studies of the physicomechanical characteristics revealed that the total coating thickness ranged from 4 to 7 µm, while the individual layer thickness varied between 1.5 and 5.5 µm. The microhardness increased by 10 ÷ 16%, with an increase in the top-layer thickness. As a result, the tool life improved by 2.4 ÷ 3.2 times compared with the standard TiN coating, and a significant reduction in wear rate was observed.

These findings enable the development of more efficient cutting tools with an extended service life and enhanced productivity.

In study [210], a multilayer ion-plasma deposition method was proposed for forming corrosion- and wear-resistant coatings on the surfaces of nickel- and steel-based alloys. Each layer in such coatings can consist of one or more metals—titanium, zirconium, molybdenum, tungsten, nickel, cobalt, iron, chromium, or aluminum—as well as solid solutions or interstitial phases based on these elements.

This method is suitable for compressor blades, aircraft engine components, gas pumping system elements, and power plant components.

Corrosion-resistance tests on compressor blades revealed that the proposed multilayer coating provided a substantial improvement in pitting corrosion resistance:180 pits were observed on uncoated blades,150 pits on blades with a standard TiN coating,No localized corrosion on blades coated using the proposed method.

Erosion-resistance testing was carried out under the following conditions: flow velocities of 120 ÷ 150 m/s, abrasive feed rate of 50 g/min, and impact angles ranging from 8° to 90°. The results showed that the fatigue strength of the blades remained practically unchanged after coating deposition, ensuring the required reliability of the components.

Experimental data indicate that the number of coating layers can vary from 5 to 300, depending on the operating conditions. The optimal range was found to be 5–30 layers—with as few as five layers being sufficient to achieve high corrosion resistance, while additional alternating wear-resistant layers can be applied for enhanced durability.

Overall, the proposed approach provides a significant increase in both corrosion and erosion resistance while maintaining high mechanical integrity, making it highly promising for use in advanced engineering applications.

The titanium alloy Ti–6Al–4V combines high strength with low density, making it an essential material in modern technology. However, its machinability is challenging, requiring advanced cutting tools and cooling techniques.

In study [211], new multilayer coatings were developed and evaluated for cutting tools used in machining Ti–6Al–4V under cryogenic cooling conditions. The coatings were deposited using PVD.

Coating M2:Consisted of four pairs of CrN/AlCrN layers with a total thickness of 2.44 µm.The structure featured decreasing layer thicknesses.Hardness: 18 GPa; Elastic modulus: 487 GPa; Porosity: 8.48%; Critical load: 47 N.

Coating M4:Consisted of eight pairs of CrN/AlCrN layers with a total thickness of 2.55 µm.Layers had uniform thickness.Hardness: 24.8 GPa; Elastic modulus: 670 GPa; Porosity: 2.09%; Critical load: 52 N.

Results:M2 exhibited the best wear resistance under cryogenic cooling, as its graded structure is optimized for extreme operating conditions.M4 demonstrated higher thermal stability and improved hardness and elasticity, making it suitable for applications requiring high mechanical and thermal resilience.

In conclusion, the PVD multilayer deposition method enabled the formation of coatings with optimized mechanical and thermophysical properties, tailored for cryogenic machining of Ti–6Al–4V.

## 4. Analysis of Bilayer Coating Systems

### 4.1. Advantages of Bilayer Coatings in Comparison with Single-Layer

In this review, bilayer coating systems represent an innovative approach to surface protection that combines the advantages of two different coating technologies such as electroplating and chemical catalytic metallization, various types of conversion coatings (thermal, chemical, electrochemical), various methods of chemical and thermal treatment, hydrophobization, and vacuum arc spraying. Their ability to solve complex problems makes duplex coatings very important. The synergistic effect of the two layers provides comprehensive protection of the bilayer coating system:Improved adhesion because of the intermediate layer;Increased corrosion resistance owing to the combined action of the layers;Wear resistance exceeds that of single-layer equivalents;Sealing the pores of the base layer with a top coat;Adaptability to specific operating conditions.

Bilayer systems of coatings show a clear edge over single-layer systems in aggressive atmospheres (chemical industry, marine environment); in conditions of combined exposure (corrosion + wear); in an area subjected to high mechanical loads (industrial equipment); and in an environment requiring tightness and durability.

The use of bilayer systems of coatings requires high initial costs compared to single-layer coatings, however, these costs are recouped by:Increased durability of the coating, which reduces the frequency of repairs;Improved protection against corrosion and mechanical damage;Saving on maintenance in the long run;Reduced equipment downtime due to a longer service life;Optimization of the cost of replacement parts;Increased operational efficiency in difficult conditions;Reducing the overall life cycle costs of the product by using less expensive materials for the base layer;Improved operational characteristics that allow the equipment to be used in extreme conditions;Application possibilities in a wider range of operating conditions;Increased reliability of protected structures.

The future development of bilayer coating systems are applied to the further improvement of application technologies, optimization of formulations, and expansion of the field of application. This will further improve the protection of materials and reduce production costs.

Thus, bilayer coating systems represent a progressive solution in material protection, combining high protective characteristics with the economic feasibility of the application.

Successful adaptation of two-layer systems for mass production requires solving technical and organizational problems, such as ensuring uniformity of the coating, quality control, scaling of processes, and standardization of materials. Examples from various industries demonstrate how these systems find application in industry:PEO and oxide coatings’ hydrophobization protects against corrosion in shipbuilding, aircraft, oil and gas, and other industries needing moisture/salt resistance [212,213];The additional coloring of anodized coatings improves the decorative effect, anti-corrosion characteristics, wear, ultraviolet radiation, and fingerprints, so it is used in the automotive industry and in the manufacture of household appliances [214];HVOF/PVD duplex coatings are being adopted in mass production to enhance the wear, corrosion, and load capabilities of components in aerospace, oil and gas, mechanical engineering, and energy [215,216,217,218].

These examples demonstrate that proper methods allow for bilayer coating systems to mass-produce across industries, resolving technical and organizational issues.

### 4.2. Analysis of the Failure Mechanisms of Bilayer Systems

From the presented data, it is possible to identify the following major forms of the destruction of two-layer systems:The formation of cracks in the upper layer because of the mismatch of the mechanical properties of the layers;Peeling of the coating from the substrate because of inadequate adhesion between the layers;Porosity of the coating, leading to the penetration of aggressive media;Deformation under thermal loads due to different expansion coefficients of the layers.

The principal causes leading to the forms of destruction of the bilayer coating systems described above are as follows:Incompatibility of the layer materials in terms of physical and mechanical properties;Incorrect technology of applying intermediate layers;Corrosion of the outer layers promotes the penetration of the electrolyte to the porous functional sublayer;Mechanical stresses during heat treatment.

Follow the following recommendations to eliminate such forms of destruction of the bilayer coating systems:Pretreatment of the surface is a critical step to create an optimal contact surface and achieve maximum adhesive strength of the joint;Use additional heat treatment to relieve internal stresses, as well as optimize the composition of the sublayers to improve adhesion;Control the parameters of the application process of each layer. It is necessary to carefully consider the selection of intermediate layers, taking into account the gradual changes in properties from the substrate to the outer coating;Apply consistent treatment with organic compounds to seal pores or use hydrophobizers to protect against moisture penetration in the top layer of the coating.

Using the right materials and technology to apply bilayer systems will significantly improve their performance and service life. The key factor is the careful design of the coating structure, taking into account all possible mechanisms of destruction.

### 4.3. Economic Efficiency and Scalability of Industrial Functional Sublayer Production

This review examined functional sublayers produced via electroplating, chemically catalytic metallization, and various conversion coatings (thermal, chemical, electrochemical). As noted in Section 2, employing these coatings as functional sublayers is unconventional. However, their inherent adhesive, strength, and anti-corrosion properties warrant their consideration for this role. Industrial-scale implementation of these coatings presents both advantages and disadvantages, including:Conversion intermediate layers exhibit excellent adhesion to a variety of metal substrates. The production process is economically advantageous, featuring optimized energy consumption and a streamlined technological sequence. Scaling this conversion coating technology for any production volume is straightforward and maintains quality. Optimizing conversion coating production maximizes economic efficiency, particularly in large-scale manufacturing.Intermediate layers of ni–p coatings exhibit excellent mechanical bonding strength and high-temperature stability. The equipment for applying these coatings is more cost-effective to purchase and operate. Ni–p coating processes offer significant flexibility in scaling production and can be easily adapted to any volume. Production capacity is most efficiently utilized with average output volumes.Intermediate layers of PEO coatings exhibit significant surface roughness, which, on the one hand, reduces corrosion resistance in aggressive environments, and on the other hand, leads to wear resistance due to microrelief, as well as improved adhesion to subsequent layers. The process of forming intermediate layers in the form of PEO coatings is accompanied by significant energy consumption. The production line for applying PEO coatings requires serious investments in technological equipment. It is possible to scale production facilities based on PEO technology only if the technological parameters are strictly observed and the process is constantly monitored. The production potential of PEO coating technology is revealed most fully in the conditions of small-scale production.

Thus, the determination of the appropriate technology must be based on a comprehensive analysis of the production environment. It is important to carry out a comprehensive calculation of the economic indicators and assess opportunities for the further optimization and modernization of production.

## 5. Conclusions

The reviewed approaches demonstrate that bilayer coatings, formed through the use of functional sublayers and additional top layers, represent an effective tool for the targeted modification of material surfaces. Sublayers provide reliable adhesion, optimal stress distribution, and enhanced corrosion resistance, thereby creating a robust foundation for the deposition of subsequent layers. Top modifying coatings, in turn, enable the sealing of porous structures, impart superhydrophobic properties to the surface, and significantly improve wear and crack resistance.

The two-stage approach offers the flexibility to combine various technologies, including electrochemical, chemical-thermal, conversion, and physical deposition methods, thus achieving a desired set of operational characteristics without considerably complicating the manufacturing process. The versatility and adaptability of such solutions determine their applicability across diverse fields, including mechanical engineering, energy, medicine, and the aerospace industry.

Therefore, the development and implementation of combined coating deposition technologies represent a promising direction in surface engineering, enabling the creation of materials with unique properties and enhanced durability.

## Figures and Tables

**Figure 1 materials-18-05217-f001:**
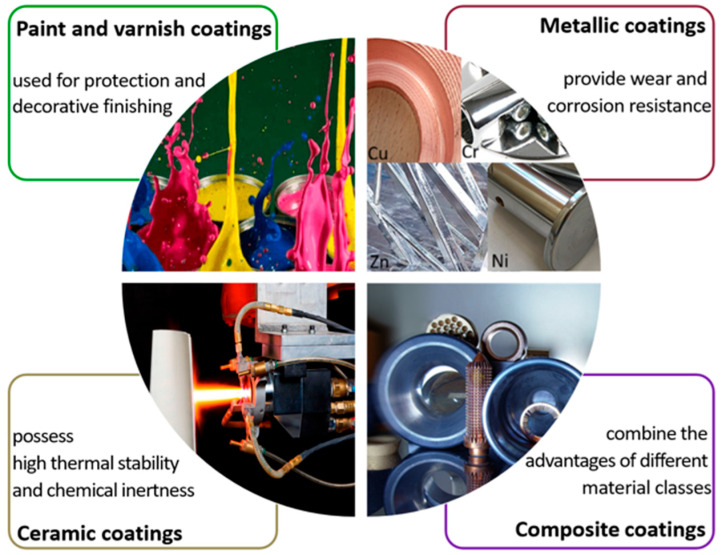
Specification of coatings by material composition and functional characteristics.

**Figure 2 materials-18-05217-f002:**
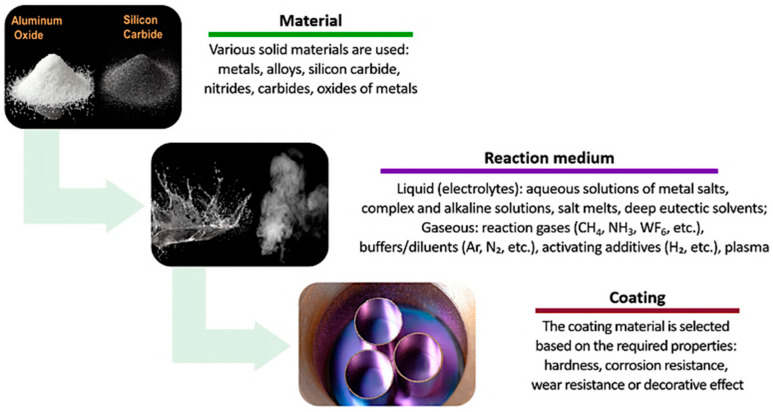
The schema explains the relationship between materials, media, and properties of the resulting coating.

**Figure 3 materials-18-05217-f003:**
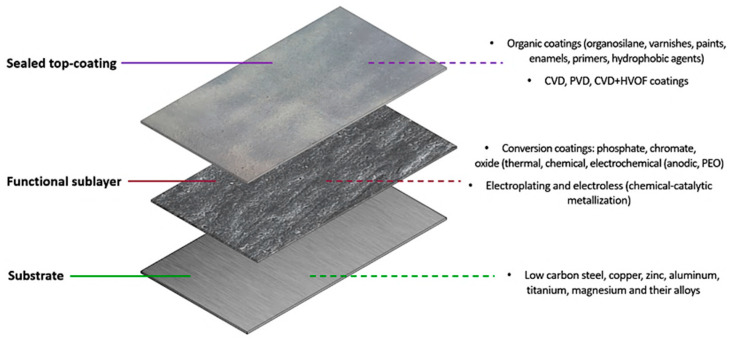
Schematic representation of the functional and sealed top-layers on the metal substrate and methods of their application considered in this review.

**Figure 4 materials-18-05217-f004:**
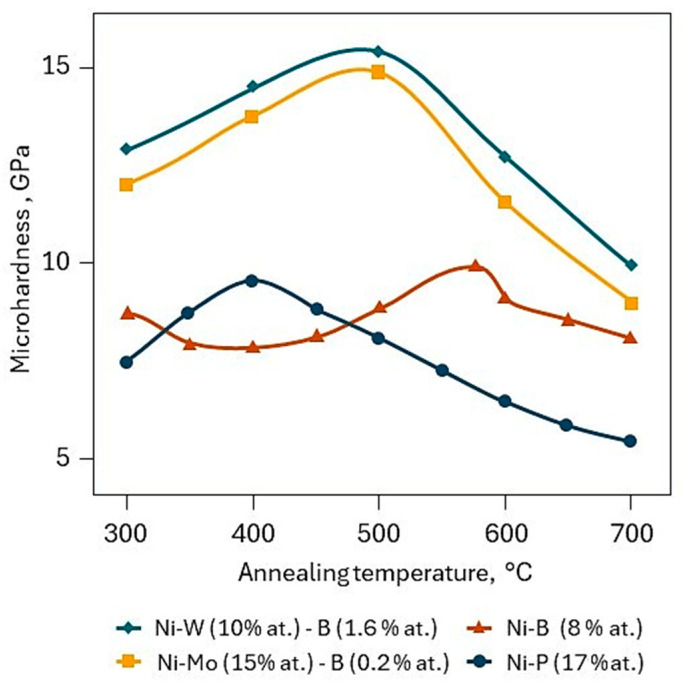
Annealing effect on the microhardness of Ni–(W,Mo)–P(B) electroless plated coatings.

**Figure 5 materials-18-05217-f005:**
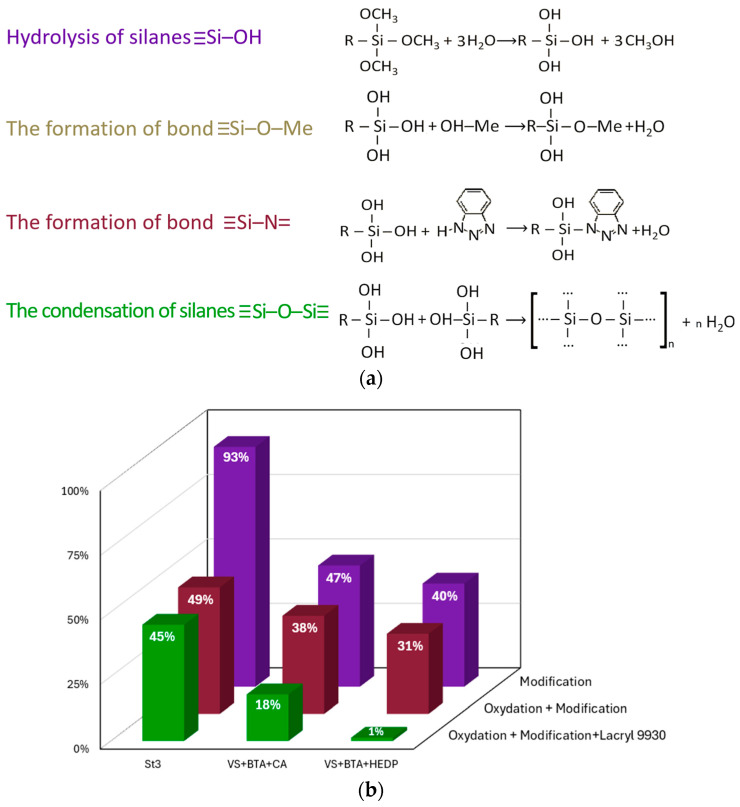
(**a**) Film formation reactions. (**b**) The results of the corrosion test (the corrosion damage, %) after exposing the samples for 96 h in a salt spray chamber.

**Figure 6 materials-18-05217-f006:**
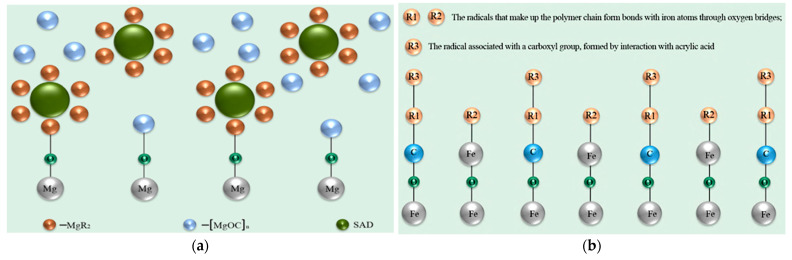
Scheme of the polymer coating on the conversion sublayer, depending on the meta substrate: (**a**) MA2; (**b**) St3.

**Figure 7 materials-18-05217-f007:**
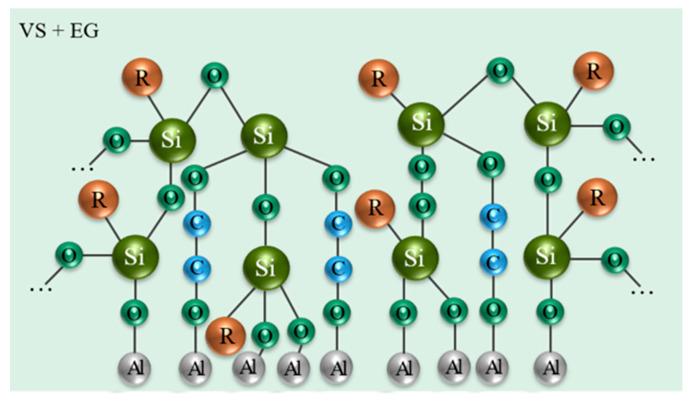
Scheme of the formation of a polymer siloxane coating during vapor-gas deposition on the surface of aluminum.

**Figure 8 materials-18-05217-f008:**
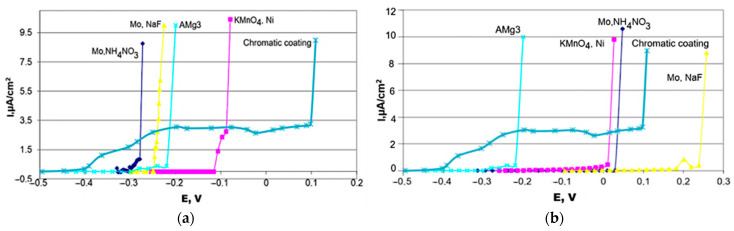
Anodic polarization curves of OCC, pure AMg-3, and chromatic coating in a solution of BB + 0.01 M NaCl, pH 7.4: (**a**) Without filling (sealing); (**b**) after filling with corrosion inhibitors.

**Figure 9 materials-18-05217-f009:**
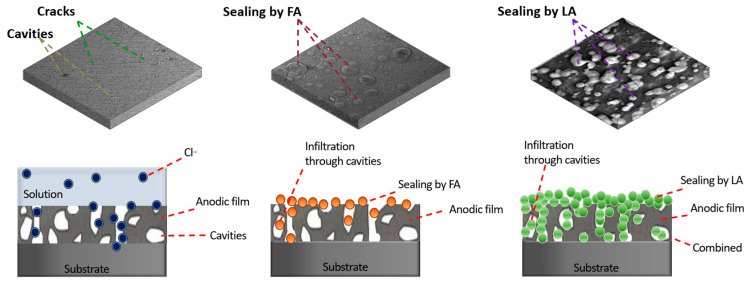
The mechanism of formation of an anticorrosive coating formed on a pre-anodized substrate with subsequent sealing with FA and LA molecules.

**Figure 10 materials-18-05217-f010:**
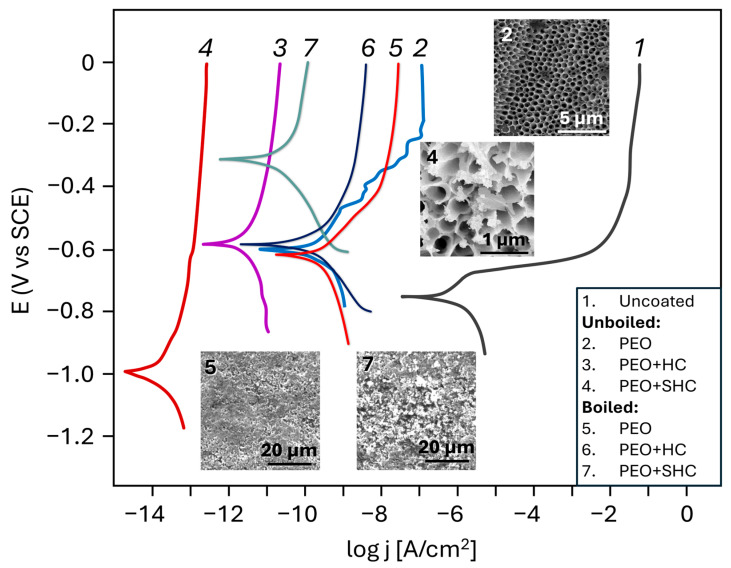
SEM images of the PEO, PEO + HC, and PEO + SHC coatings and also their potentiodynamic measurement data.

**Figure 11 materials-18-05217-f011:**
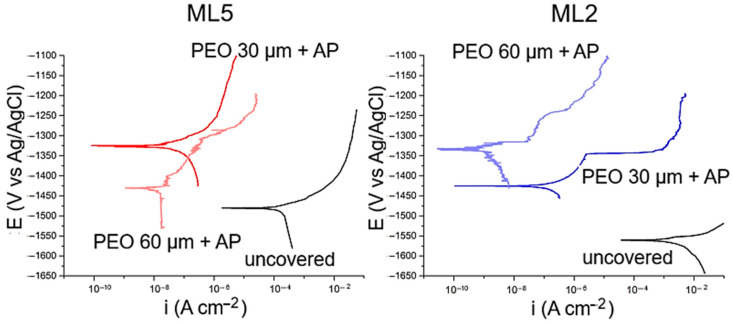
Potentiodynamic polarization curves in 3% NaCl on alloys ML5 and MA2 without coatings and with coatings PEO + AP.

**Figure 12 materials-18-05217-f012:**
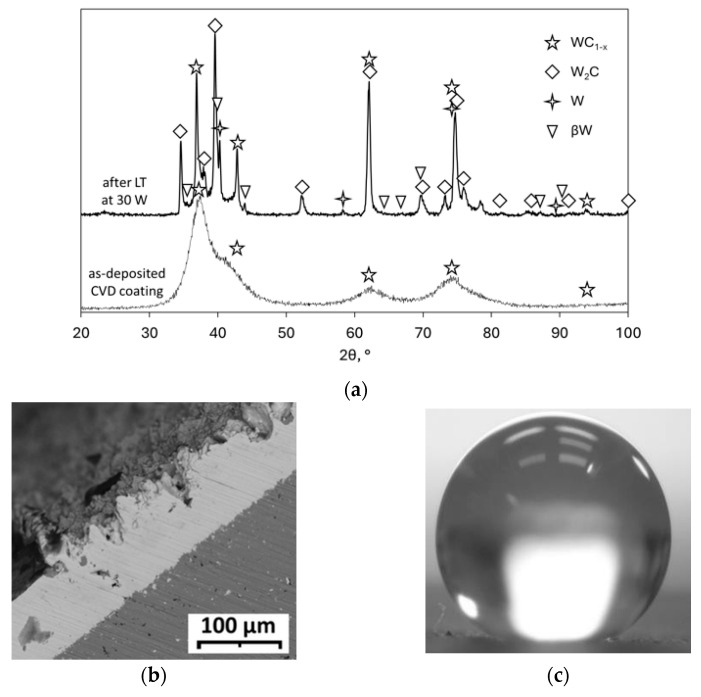
Two-stage treatment of the CVD tungsten carbide coating: (**a**) X-ray diffractograms of the coating in the initial state and after laser treatment at 30 W; (**b**) depth of the CVD layer affected by the treatment; (**c**) photograph of a droplet with a contact angle of 168° [193]. Copyright 2025 VACOR, licensed under CC-BY 4.0. https://creativecommons.org/licenses/by-nc-nd/4.0/legalcode (accessed on 25 June 2025).

**Figure 13 materials-18-05217-f013:**
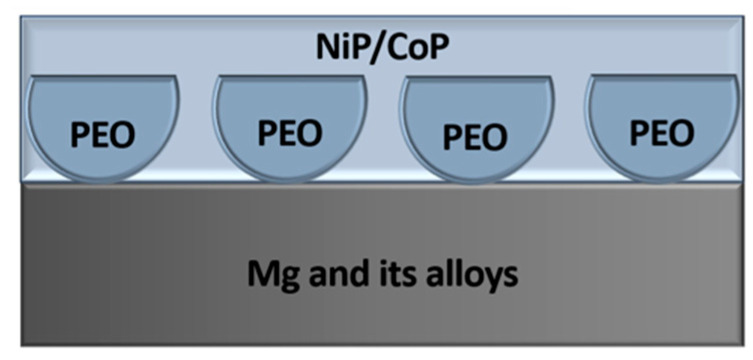
Schematic representation of a magnesium substrate with a pre-deposited PEO sublayer with a deposited continuous layer of nickel-phosphorus or cobalt-phosphorus alloys.

## Data Availability

No new data were created or analyzed in this study. Data sharing is not applicable to this article.

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
