# Peer review of "Bilayer Coating Systems: Functional Interlayers and Top Layers for Enhanced Performance"

_materials, 2025, doi:10.3390/ma18225217_

Round 1

Reviewer 1 Report

Comments and Suggestions for Authors

The article submitted for review is a comprehensive review of the literature on double-layer coatings. I suggest its publication after reviewing the following comments:
- the classification of coatings should be presented graphically, followed by a description. The same approach should be taken with the description of coating technologies. This will significantly improve the readability of the article. 
- There is no explanation as to why the topic of double-layer coatings was chosen and how they compare to other coatings. Why are they more popular than others and what is the basis for their superior performance and functionality compared to intermediate coatings, as suggested in the title of the article? 
- It is worth giving specific examples of the use of anti-wear coatings, e.g. in railways, aviation, medical engineering, press-fit connections, etc., and supporting them with relevant literature.
- As a supplement to the introduction to point 2, it is worth presenting the method of applying coatings with an interlayer using diagrams. 
- The content of the article should be organised according to a certain scheme. For example, according to the manufacturing technology (CVD, PVD) or properties. This division should be clear and legible. 

Reviewer 2 Report

Comments and Suggestions for Authors

This review comprehensively summarizes bilayer coating systems, covering functional interlayers (e.g., Ni–P, conversion coatings, PEO) and top layers (e.g., organic sealing, duplex coatings) for enhanced adhesion, corrosion resistance, and mechanical performance. It synthesizes diverse fabrication techniques (electroplating, CVD, thermal spray, PEO) and applications across aerospace, automotive, and biomedical fields, providing valuable insights for researchers and engineers. However, critical gaps exist: no systematic comparative analysis of coating performance under standardized conditions is provided; environmental and economic trade-offs of different bilayer systems are overlooked; and emerging trends (e.g., self-healing, smart coatings) are insufficiently addressed. The review also lacks a unifying framework to guide coating selection based on substrate type or operating environment, limiting its practical utility. While extensive, it would benefit from stricter organization, quantitative data synthesis, and critical evaluation of limitations in existing studies to strengthen its scientific rigor.

Major Comments (10 Critical Issues)

  1. The review describes numerous bilayer systems but provides no quantitative comparison of key performance metrics (e.g., corrosion current density, adhesion strength, wear rate) across different combinations. Why is there no standardized benchmarking (e.g., in 3.5% NaCl for corrosion, Taber abraser for wear) to help readers identify optimal systems for specific applications?
  2. Environmental sustainability is a pressing concern, yet the review rarely discusses the eco-friendliness of bilayer coatings (e.g., chromate-containing conversion coatings vs. green alternatives). How do different bilayer systems compare in terms of toxic chemical usage, energy consumption, and recyclability, and what guidelines exist for selecting sustainable options?
  3. The "functional interlayers" and "top layers" sections cover diverse technologies but lack a unifying selection framework. For a given substrate (e.g., magnesium alloy) and environment (e.g., marine), how should one prioritize interlayer/top layer combinations, and what are the key decision criteria (e.g., cost, process complexity, durability)?
  4. Emerging trends like self-healing bilayer coatings or stimuli-responsive systems are only briefly mentioned. Why is there no in-depth discussion of recent advances (post-2020) in smart bilayer coatings, and how do these innovations address current limitations (e.g., poor long-term durability)?
  5. The review highlights PEO coatings as versatile interlayers but does not critically evaluate their limitations (e.g., high energy consumption, surface roughness). How do PEO-based bilayer systems compare to alternative interlayers (e.g., Ni–P, conversion coatings) in terms of cost-effectiveness and scalability for industrial production?
  6. No analysis of failure mechanisms for bilayer systems is provided. What are the most common failure modes (e.g., interlayer delamination, top layer degradation) for different bilayer combinations, and how can these be mitigated through design or process optimization?
  7. The review focuses on lab-scale performance but rarely mentions industrial-scale challenges (e.g., uniform coating on complex geometries, batch-to-batch consistency). What modifications are required to translate lab-developed bilayer systems to large-scale manufacturing, and what case studies of industrial adoption exist?
  8. Biomedical applications are cited, but biocompatibility of bilayer coatings (e.g., ion release from Cu-containing layers, cytotoxicity of organic top layers) is not addressed. How do bilayer systems meet biocompatibility standards (e.g., ISO 10993) for implantable devices, and what are the trade-offs between corrosion resistance and biocompatibility?
  9. Quantitative data on long-term durability (e.g., >1000 h salt spray, cyclic fatigue) are scattered and not synthesized. Why is there no meta-analysis of long-term performance data, and what factors (e.g., interlayer thickness, top layer sealing) most strongly influence longevity?
  10. The review mentions duplex coatings (e.g., HVOF + PVD) but does not compare their performance to single-layer or other bilayer alternatives. In what scenarios do duplex bilayer systems outperform simpler coatings, and what are the cost-benefit justifications for their complexity?

Round 2

Reviewer 1 Report

Comments and Suggestions for Authors

The manuscript has been revised according to the reviewer's suggestions. I suggest it be published.

Reviewer 2 Report

Comments and Suggestions for Authors

No more comment.